# Path following algorithms for $\ell_2$-regularized M-estimation with approximation guarantee

**Yunzhang Zhu**
Department of Statistics
Ohio State University
Columbus, OH 43015
`zhu.219@osu.edu`

**Renxiong Liu**[*]
Statistics and Data Science team
Nokia Bell Labs
Murray Hill, NJ 07974
`renxiong.liu@nokia-bell-labs.com`

## Abstract

Many modern machine learning algorithms are formulated as regularized M-estimation problems, in which a regularization (tuning) parameter controls a trade-off between model fit to the training data and model complexity. To select the "best" tuning parameter value that achieves a good trade-off, an approximated solution path needs to be computed. In practice, this is often done through selecting a grid of tuning parameter values and solving the regularized problem at the selected grid points. However, given any desired level of accuracy, it is often not clear how to choose the grid points and also how accurately one should solve the regularized problems at the selected gird points, both of which can greatly impact the overall amount of computation. In the context of $\ell_2$-regularized M-estimation problem, we propose a novel grid point selection scheme and an adaptive stopping criterion for any given optimization algorithm that produces an approximated solution path with approximation error guarantee. Theoretically, we prove that the proposed solution path can approximate the exact solution path to arbitrary level of accuracy, while saving the overall computation as much as possible. Numerical results also corroborate our theoretical analysis.

## 1 Introduction

Modern machine learning algorithms are often formulated as regularized M-minimization problems,

$$\theta(\lambda) = \arg\min_\theta L_n(\theta) + \lambda p(\theta), \tag{1}$$

where $L_n(\theta)$ denotes an empirical loss function, $p(\theta)$ denotes a regularization function, and $\lambda$ is a tuning parameter that controls the trade-off between model fit and model complexity. Varying the tuning parameter leads to a collection of models, among which one of them may be chosen as the final model. This requires solving a collection of related optimization problems, whose solutions are often referred to as the solution path (We refer readers to Figure S4 and S5 in the supplementary material for some examples of solution paths).

Very often the exact solution path $\theta(\lambda)$ can not be computed, and path-following algorithms are usually used to obtain a sequence of solutions at some selected grid points to produce an approximated solution path. Existing path-following algorithms [see, e.g., Hastie et al., 2004, 2007] typically choose a set of equally-spaced grid points (often on a log-scale), and choose certain algorithm to solve the minimization problem at the selected grid points. Warm-start strategy is often used to leverage the fact that the minimizers at two consecutive grid points may be close. However, there is a paucity of literature on how to choose these grid points and how accurately one should solve the optimization

---

[*]Work done when Renxiong Liu is a PhD student at the Ohio State University.

37th Conference on Neural Information Processing Systems (NeurIPS 2023).

problem at the selected grid points, both of which would influence the overall computational cost and approximation accuracy.

In the context of $\ell_2$-regularized M-estimation where $p(\theta) = (1/2)\|\theta\|_2^2$, we propose a novel grid point selection scheme and an associated stopping criterion at each grid point for any optimization method. The proposal is motivated by first proving that the approximation error of a solution path is governed by two important quantities, one depending on how finely the grid points are spaced and the other depending on how accurately one solves the regularized problem at the selected grid points. We refer these two quantities as interpolation error and optimization error. Given any optimization algorithm (that solves a single regularized problem), a stopping criterion can be constructed so that the optimization error is comparable to the interpolation error, which ensures that there is no wasteful computation at each selected grid point. Then a novel grid point selection scheme is constructed to balance the approximation errors over the entire solution path.

Theoretically, we first show that the approximation error of the proposed solution path can be upper bounded by the sum of the interpolation error and the optimization error. Given any optimization algorithm, we show that the proposed stopping criterion ensures that the optimization error is dominated by the interpolation error (see Theorem 1). We then establish a global approximation-error bound for the solution path (see Theorem 2). Optimizing this global approximation-error bound by using the proposed grid point selection scheme, we also prove that the proposed solution path can approximate the exact solution path to an arbitrary level of accuracy (see Theorem 3). Finally, we establish that the total number of grid points required for the proposed algorithm to achieve an $\epsilon$-suboptimality is at most $\mathcal{O}(\epsilon^{-1/2})$ (see Theorem 4).

Path-following techniques have been used in many optimization algorithms. For instance, it is used in the interior point methods, which is one of most popular algorithm to solve general convex optimization problems [see, e.g., Chapter 1.3 of Nesterov and Nemirovskii, 1993]. However, the focus is often on the limit of the path rather than the entire solution path. In statistics/machine learning literature, Osborne [1992] and Osborne et al. [2000] applied the homotopy techniques to obtain piecewise linear solution paths for quantile regression and LASSO, respectively. Later Efron et al. [2004], Hastie et al. [2004], and Rosset and Zhu [2007] used similar ideas to derive the solution path for a family of regularized M-estimation problems. Subsequent developments include Friedman et al. [2007], Hoefling [2010], Arnold and Tibshirani [2016], among many others. Path following algorithms for more general types of $\ell_1$-reguarlization are also available. For example, Tibshirani and Taylor [2011] used dual path algorithm to compute the solution path of a generalized LASSO problem. In the context of order weight $\ell_1$ model, Bao et al. [2019] proposed an efficient approximate solution path algorithm by using primal-dual method. These works often leverage the piecewise linearity of the solution path so that an exact path-following algorithm can be explicitly derived. For situations where the solution paths are not piecewise linear, a path-following algorithm based on Newton method was considered in Rosset [2004]. However, Rosset [2004] used a simple grid point selection scheme and only established pointwise closeness to the solution path. Subsequent research on situations where the solution paths are not piecewise linear includes the work of Garrigues and Ghaoui [2008], which introduced a homotopy algorithm tailored for solving the LASSO problem using online (sequential) observations, and the work of Qu et al. [2022], which approximates the piecewise smooth path in self-paced learning for generalized self-paced regularizer.

To the best of our knowledge, there is a paucity of research on the impact of grid points selection and how to set stopping criterion to save computation. One line of related works use a piecewise constant interpolation scheme to approximate the solution path to an arbitrary level of accuracy [see, e.g., Giesen et al., 2010, 2012, 2014, Ndiaye et al., 2019]. Most relevant to our work here are the works of Giesen et al. [2012] and Ndiaye et al. [2019]. Giesen et al. [2012] proposed a generic algorithm to compute a piecewise constant approximate solution path for a family of optimization problems parameterized concavely[2] in a single hyperparameter. They showed that the number of grid points needed to achieve an $\epsilon$ suboptimality is at most $\mathcal{O}(1/\sqrt{\epsilon})$. However, their proposed method relies on the existence of a problem-dependent oracle, which for any chosen hyperparameter produces both an $\epsilon$-approximate solution at that hyperparameter and a concave polynomial that is used to compute the step size. Except for special examples like standard SVM, it not clear how to implement the oracle for general problems.

---

[2]It requires that the object function is concave with respect to the hyperparameter for any fixed value of parameters. This includes most regularized machine learning problems as special cases.

In view of these limitations, Ndiaye et al. [2019] considered regularized convex optimization problems and proposed general strategies to choose grid points based on primal-dual methods. Again, a piecewise constant path is constructed to approximate the solution path. However, their approach can only be applied to a special class of loss functions that are composition of a "simple" function (whose conjugate can be analytically derived) and affine functions. Moreover, the choice of the grid points depends on some convexity and smoothness parameters associated with the loss, which either are not available except for some special class of functions or require careful tuning due to their great impact on grid points selection. Ndiaye et al. [2019] also showed that to achieve $\epsilon$ suboptimality, the number of grid points required is $\mathcal{O}(1/\sqrt[d]{\epsilon})$ for uniformly convex loss of order $d$ [Bauschke et al., 2011] and $\mathcal{O}(1/\sqrt{\epsilon})$ for generalized self-concordant functions [Sun and Tran-Dinh, 2017]. In our paper, we show that the number of grid points needed is $\mathcal{O}(1/\sqrt{\epsilon})$ for general differentiable convex loss (see Theorem 4). Compared to Ndiaye et al. [2019], we also demonstrate through simulations that our method produces better approximated solution paths under a fixed computational budget (see Section 4). Moreover, our method can also be applicable to loss functions that are possibly nonconvex (see Section 3).

We also note that Proposition 38 of Ndiaye [2018] shows that under some regularity conditions $\epsilon$-suboptimality along the entire solution path can be achieved by using a piecewise linear interploation. However, it not clear how these conditions can be used to construct an algorithm with approximation guarantee. Our proposed algorithm, on the other hand, provides an easily implementable grid point selection scheme and an associated stopping criterion at each grid point, which together ensures that $\epsilon$-suboptimality can be attained through a piecewise linear interploation.

More recently, in the context of $\ell_2$-regularized convex optimization problems, Zhu and Liu [2021] proposed two path-following algorithms by using Newton method and gradient descent method as the basis algorithm seperately, and discuss the grid point selection scheme for each algorithm based on the piecewise linear interpolation. Our paper differs from Zhu and Liu [2021] in twofold. First, the grid-point selection scheme in Zhu and Liu [2021] depends on the basis algorithm. By contrast, in our paper any basis algorithm can be applied at any chosen grid point as long as the stopping criterion is satisfied by the basis algorithm. Next, the path-following algorithms by Zhu and Liu [2021] can only be applied to convex optimization problems, while in this article we also consider loss functions that are possibly nonconvex (see Section 3).

The rest of the paper is organized as follows. Section 2 introduces the proposed path following algorithm and establish its global approximation-error bound. Section 3 considers an extension to nonconvex empirical loss. In Section 4, we compare the the proposed scheme to a standard path-following scheme through a simulated study using ridge regression and $\ell_2$-regularized logistic regression. We close the article with some remarks in Section 5. All proofs are included in the supplementary material.

## 2 Path following algorithms

We consider an $\ell_2$-regularized M-estimation problem. More specifically, denote by $L_n(\theta)$ some empirical loss function, where $\theta \in \mathbb{R}^p$ denotes a parameter vector. We consider the solution path of an $\ell_2$-regularized M-estimation problem:

$$\theta(t) = \arg\min_{\theta \in \mathbb{R}^p} \left\{ (e^t - 1) \cdot L_n(\theta) + (1/2) \cdot \|\theta\|_2^2 \right\} . \tag{2}$$

Note that as the tuning parameter $t$ varies from 0 to $\infty$, the solution $\theta(t)$ varies from $\mathbf{0}$ to a minimizer of $L_n(\theta)$. We also remark that the choice of $e^t - 1$ is not essential. In fact, it can be replaced by any function $C(t)$ that is a strictly increasing and differentiable function, with $C(0) = 0$ and $\lim_{t\to\infty} C(t) = \infty$. The solution path produced should be independent of $C(t)$. Moreover, we can show that the $\ell_2$ norm of the solutions is an increasing function of $t$, and the solution $\theta(t)$ converges to the minimum $\ell_2$ norm minimizer of $L_n(\theta)$ (if it is finite) as $t$ goes to infinity (see Corollary S1 in the supplementary material).

Suppose that the goal is to approximate the solution path $\theta(t)$ over a given interval $[0, t_{\max})$ for some $t_{\max} \in (0, \infty]$, where we allow $t_{\max} = \infty$. Given a set of grid points $0 < t_1 < \cdots < t_N < \infty$, and approximated solutions $\{\theta_k\}_{k=1}^N$ at these grid points, we construct an approximated solution path over $[0, t_{\max})$ through linear interpolation. More specifically, we define a piecewise linear solution

path $\tilde{\theta}(t)$ as follows

$$
\begin{aligned}
\tilde{\theta}(t) &= \tfrac{t_{k+1}-t}{t_{k+1}-t_k}\theta_k + \tfrac{t-t_k}{t_{k+1}-t_k}\theta_{k+1} && \text{for any } t \in [t_k, t_{k+1}], k = 0, 1, \ldots, N-1\,, \\
\tilde{\theta}(t) &= \theta_N && \text{for any } t_N < t \le t_{\max} \text{ if } t_N < t_{\max}\,,
\end{aligned}
\tag{3}
$$

where $t_0 = 0$ and $\theta_0 = \mathbf{0}$. This defines an approximated solution path $\tilde{\theta}(t)$ for any $t \in [0, t_{\max}]$, which interpolates the approximated solutions at the selected grid points. In view of this definition, we may also assume that $t_{N-1} \le t_{\max}$, because we do not need $\tilde{\theta}(t)$ over $t \in [t_{N-1}, t_N]$ if $t_{N-1} > t_{\max}$.

To assess how well $\tilde{\theta}(t)$ approximates $\theta(t)$, we use the function-value suboptimality of the solution paths defined by $\sup_{0 \le t \le t_{\max}}\{f_t(\tilde{\theta}(t)) - f_t(\theta(t))\}$, where $f_t(\theta) := (1 - e^{-t})L_n(\theta) + e^{-t}(\|\theta\|_2^2/2)$ is a scaled version of the objective function in (2). In what follows, we call $\sup_{0 \le t \le t_{\max}}\{f_t(\tilde{\theta}(t)) - f_t(\theta(t))\}$ the global approximation error of $\tilde{\theta}(t)$, and $\sup_{t_k \le t \le t_{k+1}}\{f_t(\tilde{\theta}(t)) - f_t(\theta(t))\}$ the local approximation error of $\tilde{\theta}(t)$ over $[t_k, t_{k+1}]$.

Intuitively, how well $\tilde{\theta}(t)$ approximates $\theta(t)$ depends on how finely spaced the grid points are and the quality of the solutions at the selected grid points (i.e., $\theta_k$'s). Our first main result relates the local approximation error of $\tilde{\theta}$ over $[t_k, t_{k+1}]$ to the choice of the grid points and the accuracy of the approximated solutions at the selected grid points measured by the size of the gradients $\|\nabla f_{t_k}(\theta_k)\|_2$, where we assume that $L_n(\theta)$ is differentiable.

**Theorem 1.** *Assume that $L_n(\theta)$ is a differentiable convex function. Let $g_k := \nabla f_{t_k}(\theta_k) = (1 - e^{-t_k})\nabla L_n(\theta_k) + e^{-t_k}\theta_k$ denote the scaled gradient at $\theta_k$. For any $0 < t_1 < t_2 < \cdots < t_N < \infty$, we have that*

$$
\sup_{t \in [0, t_1]} \left\{ f_t(\tilde{\theta}(t)) - f_t(\theta(t)) \right\} \le \max\left(e^{t_1}\|g_1\|_2^2,\ \|\theta_1\|_2^2\right) + \frac{e^{t_1}(1 - e^{-t_1})^2}{2}\|\nabla L_n(\mathbf{0})\|_2^2\,, \tag{4}
$$

$$
\sup_{t \in [t_k, t_{k+1}]} \left\{ f_t(\tilde{\theta}(t)) - f_t(\theta(t)) \right\} \le e^{t_{k+1}} \max\left\{ \left(\frac{1 - e^{-t_{k+1}}}{1 - e^{-t_k}}\right)^2 \|g_k\|_2^2,\ \|g_{k+1}\|_2^2 \right\}
$$
$$
+ (e^{-t_k} - e^{-t_{k+1}})^2 \max\left\{ \frac{e^{t_{k+1}}\|\theta_k\|_2^2}{(1 - e^{-t_k})^2},\ \frac{e^{t_k}\|\theta_{k+1}\|_2^2}{(1 - e^{-t_{k+1}})^2} \right\} \tag{5}
$$

*for any $k = 1, \ldots, N-1$. If we further assume that $\|\theta(t_{\max})\|_2 < \infty$, then we have that*

$$
\sup_{t_N < t \le t_{\max}} \left\{ f_t(\tilde{\theta}(t)) - f_t(\theta(t)) \right\} \le \frac{e^{t_N}}{2(1 - e^{-t_N})}\|g_N\|_2^2 + \frac{1}{2(e^{t_N} - 1)}\|\theta(t_{\max})\|_2^2 \tag{6}
$$

*when $t_N < t_{\max}$.*

The proof of Theorem 1 starts with relating the local approximation error to the suboptimality of $\theta_k$ and $\theta_{k+1}$ at some $t \in [t_k, t_{k+1}]$. The suboptimality can then further bounded by the square norm of the grandient leveraging the fact that the objective function is $e^{-t}$-strongly convex. Finally, a triangular inequality is used to bound the norm of the grandient by quantities that depend on $\|g_k\|_2$ and $\|\theta_k\|_2$.

We can see from Theorem 1 that the upper bounds for the local approximation error in (4)–(6) consist of two terms, with the first one (depending on $g_k$) being algorithm dependent and the other term stemming from interpolation. We call them *optimization error* and *interpolation error*, respectively. Note that the optimization error is roughly of order $e^{t_k}\|g_k\|_2^2$, which again depends on how accurate the solutions $\theta_k$ are at the selected grid points. And the interpolation error is essentially independent of the quality of the solutions $\theta_k$'s, as the interpolation error only depends on the spacing of the grid points and the norm of the solutions along the solution path (typically $\|\theta_k\|_2 = \mathcal{O}(\|\theta(t_k)\|_2)$, c.f., Lemma S2 in the supplementary material). In other words, the interpolation error is irreducible once the grid points are chosen, while the optimization error does depend on the algorithm and it can be pushed to be arbitrarily small if we run the algorithm long enough at each grid point. In this sense, if the goal is to minimize the local approximation error, then both the grid points and optimization algorithm should be designed carefully to strike a balance between these two errors to minimize the overall computations. For instance, it would be wasteful to have the optimization error smaller than the interpolation error, because the additional computation would not improve the overall approximation error (at least in order).

Motivated by this observation and the local approximation-error bounds in Theorem 1, we propose a novel stopping criterion scheme at the selected grid points for any general optimization algorithm. In particular, for any given algorithm that takes an initializer and solves the problem at grid point $t_{k+1}$, we can run the algorithm initialized at $\theta_k$ until the optimization error is smaller than the interpolation error for $\theta_{k+1}$, that is

$$\underbrace{e^{t_{k+1}} \max \left\{ \left( \frac{1 - e^{-t_{k+1}}}{1 - e^{-t_k}} \right)^2 \|\nabla f_{t_k}(\theta_k)\|_2^2, \ \|\nabla f_{t_k}(\theta_{k+1})\|_2^2 \right\}}_{\text{optimization error}}$$

$$\lesssim \underbrace{(e^{-t_k} - e^{-t_{k+1}})^2 \max \left\{ \frac{e^{t_{k+1}} \|\theta_k\|_2^2}{(1 - e^{-t_k})^2}, \ \frac{e^{t_k} \|\theta_{k+1}\|_2^2}{(1 - e^{-t_{k+1}})^2} \right\}}_{\text{interpolation error}}, \quad (7)$$

where the LHS is the optimization error and the RHS is the interpolation error in the bounds in Theorem 1. This condition can be further simplified, under which the upper bounds in Theorem 1 can also be further simplified and combined so that we can bound the global approximation error of $\tilde{\theta}(t)$. This is established in the following theorem.

**Theorem 2.** *Assume that $L_n(\theta)$ is a differentiable and convex. Suppose that $2^{-1}\alpha_k \leq \alpha_{k+1} \leq 2\alpha_k$, $\alpha_k \leq \alpha_{\max}$ for some $\alpha_{\max} \leq \ln(2)$, and*

$$\|\nabla f_{t_k}(\theta_k)\|_2 \leq C_0 \frac{(e^{\alpha_k} - 1)}{(e^{t_k} - 1)} \|\theta_k\|_2; \ 1 \leq k \leq N \quad (8)$$

*for some $C_0 \leq 1/4$. Denote*

$$A = 4(1 + C_0^2(e^{\alpha_{\max}} + e^{\alpha_{\max}/2} + 1)^2), \quad B = (e^{\alpha_1} - 1)^2 \|\nabla L_n(\mathbf{0})\|_2^2,$$

$$C = \max_{1 \leq k \leq N} \frac{e^{-t_k} \|\theta_k\|_2^2 (e^{\alpha_{k+1}} - 1)^2}{(1 - e^{-t_k})^2}, \quad D = \frac{1 - e^{-\max(\alpha_N, t_{\max} - t_N)}}{e^{t_N} - 1} \max\{\|\theta_N\|_2^2, \|\theta(t_{\max})\|_2^2\}.$$

*Then*

$$\sup_{0 \leq t \leq t_{\max}} \{f_t(\tilde{\theta}(t)) - f_t(\theta(t))\} \leq A \max\{B, C, D\} \quad (9)$$

*when $t_N < t_{\max}$ and $\|\theta(t_{\max})\|_2 < \infty$, and*

$$\sup_{0 \leq t \leq t_{\max}} \{f_t(\tilde{\theta}(t)) - f_t(\theta(t))\} \leq A \max\{B, C\} \quad (10)$$

*when $t_{N-1} \leq t_{\max} \leq t_N$ for some $N \geq 1$.*

The above result provides a practical approach to verify whether the optimization error is dominated by the interpolation error by checking (8). In other words, (8) is a sufficient condition for (7) and it can be used as a stopping criterion at each grid point $t_k$ for any algorithm that computes a solution at $t_k$. We also remark that, using the above bounds, similar approximation-error bounds for $\|\tilde{\theta}(t) - \theta(t)\|_2$ can be derived as well (see Corollary S2 in the supplementary material).

Another important implication of Theorem 2 is that the established global approximation-error bound provides a principled approach to choose the grid points so that for any $\epsilon > 0$ and $t_{\max} \in (0, \infty]$ we have

$$\sup_{0 \leq t \leq t_{\max}} \{f_t(\tilde{\theta}(t)) - f_t(\theta(t))\} \lesssim \epsilon \quad (11)$$

for the solution path generated by an algorithm, and at the same time it tries to maximize the speed at which the algorithm explores the solution path from 0 to $t_{\max}$. In particular, to ensure (11), we propose to use the following step sizes

$$\alpha_1 = \min\{\alpha_{\max}, \ \ln(1 + \frac{\sqrt{\epsilon}}{\|\nabla L_n(\mathbf{0})\|_2})\} \quad (12)$$

and

$$\alpha_{k+1} = \min \left\{ \ln(1 + \frac{c_1(e^{\alpha_1} - 1)\|\nabla L_n(\mathbf{0})\|_2 e^{t_k/2}(1 - e^{-t_k})}{\|\theta_k\|_2}), \ \alpha_{\max}, \ 2\alpha_k \right\}, \quad (13)$$

---

**Algorithm 1** A general path following algorithm.

---

**Input**: $\epsilon > 0$, $C_0 \leq 1/4$, $c_1 \geq 1$, $c_2 > 0$, $0 < \alpha_{\max} \leq 5^{-1}$ and $t_{\max} \in (0, \infty]$.

**Output**: grid points $\{t_k\}_{k=1}^N$ and an approximated solution path $\tilde{\theta}(t)$.

1: **Initialize:** $k = 1$.
2: Compute $\alpha_1$ using (12). Starting from $\mathbf{0}$, iteratively calculate $\theta_1$ by minimizing $f_{t_1}(\theta)$ until (8) is satisfied for $k = 1$.
3: **while** (14) is not satisfied, **do**
4:     Compute $\alpha_{k+1}$ using (13), update $t_{k+1} = t_k + \alpha_{k+1}$.
5:     Starting from $\theta_k$, iteratively compute $\theta_{k+1}$ by minimizing $f_{t_{k+1}}(\theta)$ until (8) is satisfied.
6:     Update $k = k + 1$.
7: **Interpolation:** construct a solution path $\tilde{\theta}(t)$ through linear interpolation of $\{\theta_k\}_{k=1}^N$ using (3).

---

and we terminate the algorithm at $t_N$ when

$$\frac{c_2 \left(1 - e^{-\max(\alpha_N, t_{\max} - t_N)}\right)}{e^{t_N} - 1} \leq \epsilon \text{ or } t_N \geq t_{\max}, \tag{14}$$

where $c_1 \geq 1$, $c_2 > 0$, and $0 < \alpha_{\max} \leq 1/5$ are absolute constants. Note that the above step sizes and termination criterion are chosen to ensure that the upper bounds in (9) and (10) are at most $\mathcal{O}(\epsilon)$. Formally, we show that, given any $\epsilon > 0$, the above scheme achieves $\epsilon$-suboptimality (11) for $\tilde{\theta}(t)$ (up to a multiplicative constant).

**Theorem 3.** *Suppose that $L_n(\theta)$ is differentiable and convex. For any $\epsilon > 0$ and $t_{\max} \in (0, \infty]$, assume that either $t_{\max} < \infty$ or $\|\theta(t_{\max})\|_2 < \infty$. Then, using the step sizes and the termination criterion specified above in* (12), (13), *and* (14), *any algorithm that produces iterates satisfying* (8) *for any $k \geq 1$ terminates after finite number of iterations, and when terminated, the solution path $\tilde{\theta}(t)$ satisfies*

$$\sup_{0 \leq t \leq t_{\max}} \{f_t(\tilde{\theta}(t)) - f_t(\theta(t))\} \lesssim \epsilon. \tag{15}$$

Note that when $\epsilon$ is small enough, we have that $\alpha_1 = \ln\left(1 + \sqrt{\epsilon}/\|\nabla L_n(\mathbf{0})\|_2\right)$, which together with (15) implies that

$$\sup_{0 \leq t \leq t_{\max}} \{f_t(\tilde{\theta}(t)) - f_t(\theta(t))\} \lesssim (e^{\alpha_1} - 1)^2 \|\nabla L_n(\mathbf{0})\|_2^2.$$

As a result, the approximation error for the solution path is essentially controlled by the initial step size $\alpha_1$. Smaller $\alpha_1$ leads to better approximation. However, smaller $\alpha_1$ also leads to a slower exploration of the solution path and a more stringent stopping criterion at individual grid points (c.f. (8)). As such, the initial step size controls the trade-off between computational cost and accuracy. Moreover, given $\alpha_1$, we can see from (13) that how fast the algorithm can explore the solution path depends largely on $\|\theta_k\|_2$. In particular, if $\|\theta_k\|_2$ is bounded (e.g., when $\theta^\star$ is finite), then the first term in the min function of (13) increases as $t_k$ increases. Therefore, aggressive step sizes can be taken in this case until it reaches $\mathcal{O}(1)$, which will likely lead to a fast exploration of the solution path. On the other hand, if $\|\theta_k\|_2$ grows quite quickly to infinity as $k$ increases, then the first term in the min function may go to zero as $k \to \infty$ (say, when $\|\theta_k\|_2/e^{t_k/2} \to \infty$). This means that the step sizes need to decrease to zero eventually, leading to a slower exploration of the solution path.

We summarize the proposed scheme in Algorithm 1. Again, we emphasize that the above scheme can be applied to any optimization algorithm. Later, we will empirically investigate its performance using Newton method and gradient descent method.

Lastly, we establish an upper bound on the total number of grid points required for Algorithm 1 to achieve $\epsilon$-suboptimality.

**Theorem 4.** *Under the assumptions in Theorem 3, the total number of grid points required for Algorithm 1 to achieve an $\epsilon$-suboptimality* (15) *is at most $\mathcal{O}(\epsilon^{-1/2})$.*

# 3 Extensions to nonconvex loss functions

So far we assume that the empirical loss is convex. Next, we consider a generalization, where we do not assume convexity for $L_n(\theta)$. Instead, we assume that there exists $\lambda(t) > 0$ so that

$$f_t(\theta) - f_t(\theta(t)) \leq \frac{1}{2\lambda(t)} \|\nabla f_t(\theta)\|_2^2, \tag{16}$$

for any $t \geq 0$ and $\theta \in \mathbf{dom}(L_n)$. In the literature, the condition (16) was often referred to as the Polyak-Łojasiewicz (PL) inequality. Polyak [1964] proved the linear rate of convergence for gradient descent method under this condition. Also see Karimi et al. [2016] for discussions of more recent development. Note that if $L_n(\theta)$ is convex, then (16) holds for $\lambda(t) = e^{-t}$ since $f_t(\theta)$ is $e^{-t}$-strongly convex. In general, however, $L_n(\theta)$ does not need to be convex for (16) to hold.

For some technical reasons, we need to consider slightly different interpolation scheme

$$\begin{aligned}
\tilde{\theta}(t) &= \theta_k \quad \text{for any } t \in [t_k, t_{k+1}], 0 \leq k \leq N-1; \\
\tilde{\theta}(t) &= \theta_N \quad \text{for any } t_N < t \leq t_{\max} \text{ if } t_N < t_{\max},
\end{aligned} \tag{17}$$

where $t_0 = 0$ and $\theta_0 = \mathbf{0}$. As we can see that the approximated solution path is piecewise constant. We use the following termination criterion at each $t_k$

$$\begin{aligned}
\|g_1\|_2 &\leq \min(C_0(e^{\alpha_1} - 1)\|\nabla L_n(\mathbf{0})\|_2, 1); \\
\|g_{k+1}\|_2 &\leq \min(C_0 \frac{e^{\alpha_{k+1}} - 1}{e^{t_k} - 1}\|\theta_k\|_2, 1), \ k \geq 1.
\end{aligned} \tag{18}$$

We also define a slightly new step size scheme as follows:

$$\alpha_1 = \min\{\alpha_{\max}, \ln(1 + \frac{\sqrt{\epsilon \min(\lambda_0, \lambda_1)}}{\|\nabla L_n(\mathbf{0})\|_2})\} \tag{19}$$

and

$$\alpha_{k+1} = \min\{\alpha_{\max}, 2\alpha_k, \ln(1 + \frac{c_1(e^{\alpha_1} - 1)\|\nabla L_n(\mathbf{0})\|_2(e^{t_k} - 1)\sqrt{\min(\lambda_k, \lambda_{k+1})}}{\|\theta_k\|_2\sqrt{\min(\lambda_0, \lambda_1)}})\}, \tag{20}$$

where $\lambda_k = \inf_{t_k \leq t \leq t_{k+1}} \lambda(t), k \geq 0$. Finally, the stopping criterion for the path-following algorithm is set to be

$$\frac{c_3}{e^{t_N} - 1} \leq \epsilon \text{ or } t_N \geq t_{\max}, \tag{21}$$

where $c_3 > 0$ is an absolute constant.

Moreover, we need to make some additional assumptions on the loss function.

**Assumption (A1).** Assume that $L_n(\theta)$ is differentiable and $\inf_\theta L_n(\theta) > -\infty$. Moreover, assume that there exists a positive decreasing function $g(\cdot)$ such that $\lambda(t) \geq g(t) > 0$ for any $0 \leq t < \infty$.

**Theorem 5.** *Suppose that Assumption (A1) holds. For any $\epsilon > 0$ and $t_{\max} \in (0, \infty]$, assume that either $t_{\max} < \infty$ or $\sup_{0 < t \leq t_{\max}} \|\theta(t)\|_2 < \infty$. Then, using step sizes and termination criterion specified in (19), (20), and (21), any algorithm that produces gradients satisfying (18) terminates after finite number of iterations, and when terminated, the solution path $\tilde{\theta}(t)$ defined in (17) satisfies*

$$\sup_{0 \leq t \leq t_{\max}} \{f_t(\tilde{\theta}(t)) - f_t(\theta(t))\} \lesssim \epsilon. \tag{22}$$

The above result can be viewed as a generalization of Theorem 3, because when $L_n(\theta)$ is convex, we could set $\lambda(t) = \exp(-t)$ so that both condition (16) and Assumption (A1) are satisfied.

# 4 Numerical studies

In this section, we study the operating characteristics of Newton method, gradient descent method and a mixed method in ridge regression and $\ell_2$-regularized logistic regression, using both the simulated datasets and a real data example. Here the mixed method applies gradient descent method to minimize

$f_{t_k}(\theta)$ at the beginning and then switch to Newton method when the number of gradient steps $n_k$ exceeds $\min\{n, p\}$. To illustrate the advantages of the proposed grid point selection scheme and stopping criterion, we use a standard implementation of a typical path-following algorithm as a baseline. In particular, the grid points are equally spaced (on a log-scale) for both examples. For $\ell_2$ logistic regression, Newton method is used as the basis algorithm with a fixed stopping criterion $\|\nabla f_{t_k}(\theta_k)\|_2 < 10^{-5}$, while for ridge regression the exact solution can be computed. We refer to this method as the baseline method.

We also compare the proposed methods against the method of Ndiaye et al. [2019]. In particular, we compare our grid points selection scheme (13) with the so-called adaptive unilateral scheme proposed in Ndiaye et al. [2019], where Newton method is used as the basis algorithm at each grid point. Different from the aforementioned methods, adaptive unilateral scheme constructs piecewise constant solution path $\tilde{\theta}(\lambda)$ over chosen grid points to approximate the solution path $\theta(\lambda) = \arg\min_{\theta \in \mathbb{R}^p} P_\lambda(\theta)$, where $P_\lambda(\theta) = L_n(\theta) + (\lambda/2) \cdot \|\theta\|_2^2$, such that for all $\lambda \in (\lambda_{\min}, \lambda_{\max})$

$$P_\lambda(\tilde{\theta}(\lambda)) - P_\lambda(\theta(\lambda)) < \epsilon. \tag{23}$$

We call this Ndiaye's method throughout this section.

All methods considered in the numerical studies are implemented in R using Rcpp package [Eddelbuettel et al., 2011, Eddelbuettel, 2013].

**Example 1: Ridge regression.** This example considers ridge regression over a simulated dataset. In this case, the empirical loss function is $L_n(\theta) = \|Y - X\theta\|_2^2/(2n)$, where $X \in \mathbb{R}^{n \times p}$ and $Y \in \mathbb{R}^n$ denote the design matrix and the response vector. In the simulation, the data $(X, Y)$ are generated from the usual linear regression model $Y = X\theta^\star + \tilde{\epsilon}$, where $\tilde{\epsilon} \sim N(0, I_{n \times n})$, $\theta^\star = (1/\sqrt{p}, \ldots, 1/\sqrt{p})^\top$, and rows of $X$ are IID samples from $N(0, I_{p \times p})$. Throughout this example, we consider $n = 1000$ and $p = 500, 10000$,

**Example 2: $\ell_2$-regularized logistic regression.** This example considers $\ell_2$-regularized logistic regression over a simulated dataset. Let $X \in \mathbb{R}^n$ and $Y \in \{+1, -1\}^n$ denote the design matrix and the binary response vector. The empirical loss function for logistic regression is $L_n(\theta) = n^{-1} \sum_{i=1}^n \log(1 + \exp(-Y_i X_i^\top \theta))$. We simulate the data $(X, Y)$ from a linear discriminant analysis (LDA) model. More specifically, we sample the components of $Y$ independently from a Bernoulli distribution with $\mathbb{P}(Y_i = +1) = 1/2; i = 1, 2, \ldots, n$. Conditioned on $Y_i$, $X_i$'s are then independently drawn from $N(Y_i \mu, \sigma^2 I_{p \times p})$, where $\mu \in \mathbb{R}^p$ and $\sigma^2 > 0$. Note that $\mu$ and $\sigma^2$ controls the Bayes risk, which is $\Phi(-\|\mu\|_2/\sigma)$ under the 0/1 loss, where $\Phi(\cdot)$ is the cumulative distribution function of a standard normal random variable. Here we choose $\mu = (1/\sqrt{p}, \ldots, 1/\sqrt{p})$ and $\sigma^2 = 1$ so that the Bayes risk is $\Phi(-1)$, which is approximately 15%. Similar to Example 1, two different problem dimension are considered: $n = 1000$ and $p = 500, 10000$.

**Example 3: Real data example.** This example fits an $\ell_2$-regularized logistic regression using the *a9a* dataset from LIBSVM [Chang and Lin, 2011]. This dataset provides information about whether an adult's annual income surpasses $\$50,000$, along with some specific characteristics of the given adult. It contains a total of 32561 examples with 123 features and 1 binary response variable.

For all three examples, we use the global approximation error $\sup_{0 \le t \le t_{\max}}\{f_t(\tilde{\theta}(t)) - f_t(\theta(t))\}$ as an evaluation metric, where $\tilde{\theta}(t)$ is the linear interpolation of the iterates $\theta_k$ generated by each method. Throughout, we let $t_{\max} = 10$. Since this metric can not be calculated exactly, we approximate the global approximation error by $\max_{1 \le i \le 100}\{f_{s_i}(\tilde{\theta}(s_i)) - f_{s_i}(\theta(s_i))\}$, where $s_1, \ldots, s_{100}$ are uniformly sampled from $(0, t_{\max})$. Here we compute the solutions $\theta(s_i)$ using the CVX solver [Grant and Boyd, 2014, 2008] for $\ell_2$-regularized logistic regression, while calculating $\theta(s_i)$ explicitly for ridge regression.

We compare Newton method, gradient descent method and their mixed version against the baseline method and Ndiaye's method. To make a fair comparison with Ndiaye's method, we set $\lambda_{\min} = 1/(e^{10} - 1)$ to match $t_{\max} = 10$ in log scale and choose $\lambda_{\max} = 100$.

Figure 1 presents runtime versus global approximation error for Example 1 and 2 based on 100 replications, as we vary $\epsilon$ in (12) for our proposed method and (23) for Ndiaye's method. For the baseline method, we vary the spacing between consecutive grid points instead. Clearly the proposed Newton method outperforms the baseline method and Ndiaye's method in both low dimension and high dimension scenarios, which empirically confirms that the proposed grid point selection scheme

(c.f. (13)) is more efficient than both the standard equally-spaced grid point scheme and the adaptive unilateral scheme proposed by Ndiaye et al. [2019]. Moreover, gradient descent method performs similarly compared with the mixed method in both setups of Example 1, which suggests that switching does not happen much in those cases. However, in both problem dimensions of Example 2, we do see a difference between the mixed method and gradient descent, which is due to the fact that the switching to the Newton update does happen and it can speed up the computation by design. Lastly, as one would expect, Newton method is more efficient than both the gradient descent method and the mixed method when the problem dimension is small, but less so when problem dimension is large.

Figure 2 shows the runtime versus global approximation error tradeoff for our real data example, as we vary $\epsilon$ for our proposed methods and Ndiaye's method, and change grid point spacing for the baseline method. Again, the Newton method is more efficient than both the baseline method and the Ndiaye's method, which demonstrates the advantage of our proposed grid point selection scheme over the standard equally-spaced grid point scheme and the scheme by Ndiaye et al. [2019].

We also plot the number of iterations at each grid point for Newton method and gradient descent method in Figure S1-S3 of the supplementary material. Interestingly, for ridge regression (c.f., Figure S1), the number of iterations by gradient method first increases and then stays flat as $t_k$ grows. Newton method, however, only takes one iteration at each grid point. Moreover, the level of approximation (i.e., $\epsilon$) seems to have no impact on the number of iterations at each grid point, which is highly desirable. For the $\ell_2$-regularized logistic regression (c.f., Figure S2 and S3), the number of iterations needed increases as $t_k$ increases for gradient descent method, whereas the Newton method always requires just a constant number of iterations at each $t_k$. Again, the level of approximation (i.e., $\epsilon$) does not seem to influence the number of iterations much.

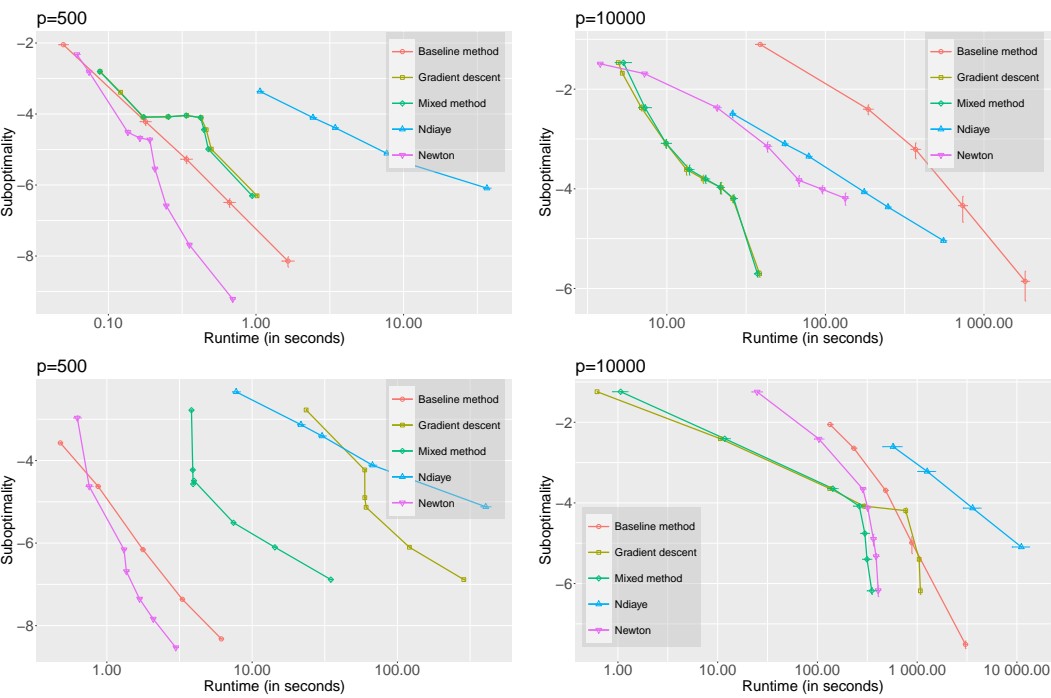

Figure 1: Runtime v.s. suboptimality for the Newton method, the gradient descent method, the mixed method, the baseline method and Ndiaye's method under two problem dimensions when applied to ridge regression (upper panels) and $\ell_2$-regularized logistic regression (lower panels). The vertical and horizontal lines at each point represent the standard errors of suboptimality and runtime.

## 5 Discussion

In this article, we proposed a novel grid point selection scheme and stopping criterion for any general path-following algorithms. A simple solution path was constructed through linear interpolation

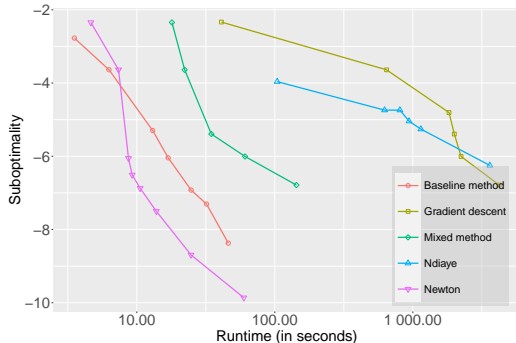

Figure 2: Runtime v.s. suboptimality for the Newton method, the gradient descent method, the mixed method, the baseline method and Ndiaye's method when applied to $\ell_2$-regularized logistic regression over the *a9a* real dataset from LIBSVM [Chang and Lin, 2011].

and theoretical approximation-error bound was derived. A generalization to a class of nonconvex empirical loss was also considered. We have not touched on the overall computational complexity analysis for commonly used optimization algorithms. Based on some preliminary analysis, we conjecture that for gradient descent method and Newton method, the number of iterations required are $\mathcal{O}(\epsilon^{-1}\ln(\epsilon^{-1}))$ and $\mathcal{O}(\epsilon^{-1/2})$, respectively. Moreover, lower bound results are also worth investigating, that is, what are the minimum number of gradient or Newton steps needed to achieve an $\epsilon$-suboptimality for the entire solution path? Moreover, our current analysis can be readily extended to general differentiable regularization functions. A more challenging future direction is to investigate the case where the empirical loss function or the regularizer are not differentiable.

## Acknowledgments and Disclosure of Funding

The authors are supported in part by the US National Science Foundation (NSF) under grant DMS-17-12580, DMS-17-21445 and DMS-20-15490. The authors thank the reviewers for valuable comments and suggestions that improved the article.

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
