# Supplementary Material for "Path following algorithms for $\ell_2$-regularized M-estimation with approximation guarantee"

**Yunzhang Zhu**
Department of Statistics
Ohio State University
Columbus, OH 43015
zhu.219@osu.edu

**Renxiong Liu**
Statistics and Data Science team
Nokia Bell Labs
Murray Hill, NJ 07974
renxiong.liu@nokia-bell-labs.com

The supplementary material collects additional numerical results in Section 4, examples of ground truth and approximated solution path, as well as all the proofs of the theorems and corollaries in the main text.

## 1 Additional numerical results in Section 4

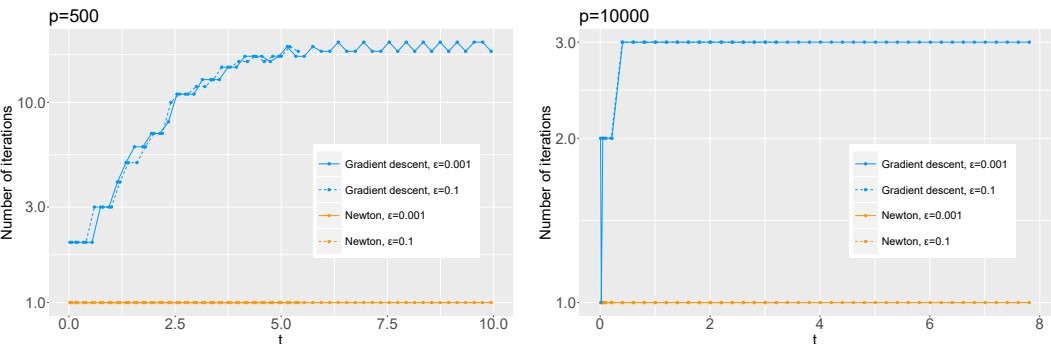

Figure S1: Number of iterations at each grid point for the Newton and gradient descent methods applying to ridge regression over simulated data generated in Example 1.

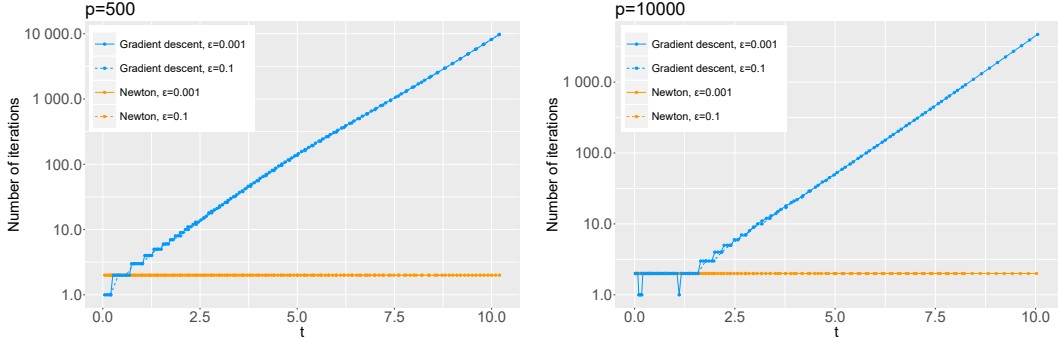

Figure S2: Number of iterations at each grid point for the Newton and gradient descent methods applying to the $\ell_2$-regularized logistic regression over simulated data generated in Example 2.

37th Conference on Neural Information Processing Systems (NeurIPS 2023).

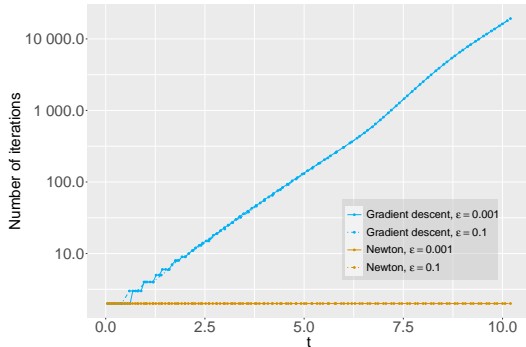

Figure S3: Number of iterations at each grid point for the Newton and gradient descent methods applying to the $\ell_2$-regularized logistic regression over the $a9a$ real dataset from LIBSVM [Chang and Lin, 2011].

In this section, we inlcude some additional numerical results for the three examples in Section 4. In particular, we plot the number of iterations at each grid point by Newton method and gradient descent method for all three examples. We summarize the results in Figure S1-S3.

Figure S1 presents the results for ridge regression. In this case, the number of iterations by gradient method first increases and then stays flat as $t_k$ grows. Newton method, however, only takes one iteration at each grid point. Moreover, the level of approximation (i.e., $\epsilon$) seems to have no impact on the number of iterations at each grid point, which is highly desirable.

Figure S2 shows the results for $\ell_2$-regularized logistic regression. As we can see from Figure S2, the number of iterations needed increases as $t_k$ increases for gradient descent method, whereas the Newton method always requires just two iterations at each $t_k$. Again, the level of approximation (i.e., $\epsilon$) does not seem to influence the number of iterations much.

Lastly, Figure S3 shows the iteration plot for the real data example. Again, similar to the results in Figure S2, for gradient descent method the number of iterations needed increases as $t_k$ increases, while the Newton method requires only a few iterations at each $t_k$.

## 2 Examples of ground truth solution path and approximated solution path

This section includes examples of visualized ground truth solution path and approximated solution path by Newton method and gradient descent method, when applied to ridge regression and $\ell_2$-regularized logistic regression.

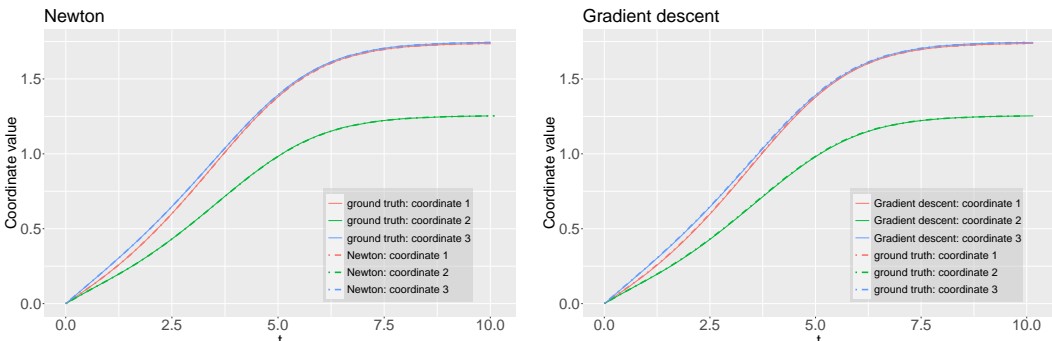

Figure S4: Ground truth solution path and approximated solution path by Newton method (left panel) and gradient descent method (right panel), when applied to ridge regression over simulated data generated in Example 1, with $n = 100$ and $p = 3$.

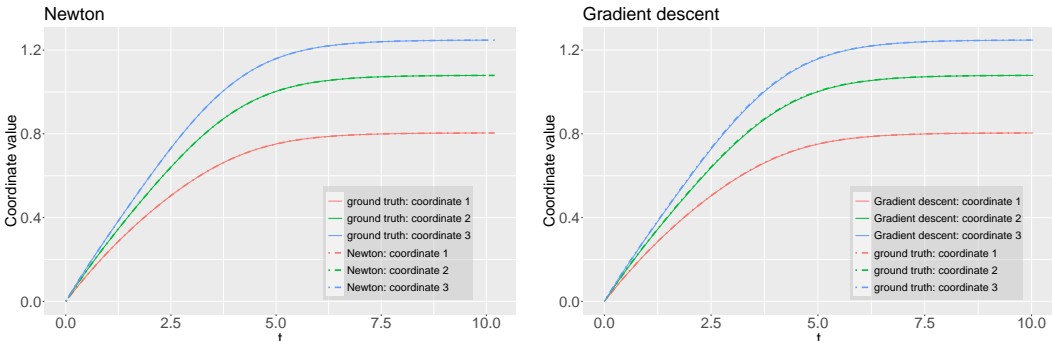

Figure S5: Ground truth solution path and approximated solution path by Newton method (left panel) and gradient descent method (right panel), when applied to $\ell_2$-regularized logistic regression over simulated data generated in Example 2, with $n = 100$ and $p = 3$.

## 3 Some preliminary results and supporting lemmas

Some standard results for $m$-strongly convex functions will be repeatedly used in the proofs, which are stated below. We omit their proofs as all of them can be found in standard convex analysis textbooks [see, e.g., Boyd and Vandenberghe, 2004]. Suppose that $f(\cdot)$ is a $m$-strongly convex function with minimizer $x^\star$. Then for any $x$ and $y$,

$$m\|x - y\|_2^2 \leq \langle \nabla f(x) - \nabla f(y), x - y \rangle \leq \frac{1}{m}\|\nabla f(x) - \nabla f(y)\|_2^2, \tag{S1}$$

$$\frac{m}{2}\|x - x^\star\|_2^2 \leq f(x) - f(x^\star) \leq \frac{1}{2m}\|\nabla f(x)\|_2^2 \text{ and } \|x - x^\star\|_2 \leq \frac{1}{m}\|\nabla f(x)\|_2 \tag{S2}$$

Next, we present two supporting lemmas.

**Lemma S1.** *Suppose that $L_n(\theta)$ is a proper convex function. Then*

$$\|\theta(t') - \theta(t)\|_2 \leq \frac{|C(t') - C(t)|}{C(t)}\|\theta(t)\|_2 \tag{S3}$$

*for any $t, t' > 0$.*

*Proof of Lemma S1.* Note that for any $t > 0$ and any $g_{t'} \in \partial L_n(\theta(t'))$ and $g_t \in \partial L_n(\theta(t))$, we have $\langle g_{t'} - g_t, \theta(t') - \theta(t) \rangle \geq 0$, where $\partial L_n(\theta)$ denotes the *subdifferential* of $L_n(\cdot)$ at $\theta$. Hence, for any $h_{t'} \in \partial f_{t'}(\theta(t')), h_t \in \partial f_{t'}(\theta(t))$

$$\langle h_{t'} - h_t, \theta(t') - \theta(t) \rangle \geq \|\theta(t') - \theta(t)\|_2^2 \tag{S4}$$

Since $\mathbf{0} \in \partial f_{t'}(\theta(t'))$ and $-C(t')\theta(t)/C(t) + \theta(t) \in \partial f_{t'}(\theta(t))$, substituting $h_{t'}$ with $\mathbf{0}$ and $h_t$ with $-C(t')\theta(t)/C(t) + \theta(t)$, we obtain that

$$\langle C(t')\theta(t)/C(t) - \theta(t), \theta(t') - \theta(t) \rangle \geq \|\theta(t') - \theta(t)\|_2^2. \tag{S5}$$

This further implies that

$$\|\theta(t') - \theta(t)\|_2^2 \leq \langle C(t')\theta(t)/C(t) - \theta(t), \theta(t') - \theta(t) \rangle \leq |C(t')/C(t) - 1|\|\theta(t)\|_2\|\theta(t') - \theta(t)\|_2,$$

which proves (S3). This completes the proof of Lemma S1. ∎

**Lemma S2.** *Assume that $L_n(\theta)$ is a closed proper convex function. We have that*

$$\|\theta(t)\|_2 \leq (e^t - 1)\|\nabla L_n(\mathbf{0})\|_2 \text{ and } \|\theta(t)\|_2 \leq (1 - e^{-t})\left(\|\nabla L_n(\mathbf{0})\|_2 + \|\theta(t')\|_2\right) \tag{S6}$$

*for any $t' \geq t > 0$. Moreover, under the assumption that (8) holds, we have that*

$$\frac{1}{1 + \frac{C_0(e^{\alpha k} - 1)}{(1 - e^{-t_k})}} \leq \frac{\|\theta_k\|_2}{\|\theta(t_k)\|_2} \leq \frac{1}{1 - \frac{C_0(e^{\alpha k} - 1)}{(1 - e^{-t_k})}} \tag{S7}$$

*for any $1 \leq k \leq N$.*

*Proof of Lemma S2.* Since $f_t(\cdot)$ is $e^{-t}$ strongly convex, using (S2), we have

$$\|\theta(t) - \mathbf{0}\|_2 \leq e^t \|\nabla f_t(\mathbf{0})\|_2 = (e^t - 1)\|\nabla L_n(\mathbf{0})\|_2 \,,$$

which proves the first inequality $\|\theta(t)\|_2 \leq (e^t - 1)\|\nabla L_n(\mathbf{0})\|_2$. Combining this with the fact that $\|\theta(t)\|_2 \leq \|\theta(t')\|_2$, we have that

$$\|\theta(t)\|_2 \leq \min\left((e^t - 1)\|\nabla L_n(\mathbf{0})\|_2, \|\theta(t')\|_2\right) \leq (1 - e^{-t})\left(\|\nabla L_n(\mathbf{0})\|_2 + \|\theta(t')\|_2\right) \,,$$

which proves the second inequality in (S6).

Next, we prove (S7). Using the fact that $\|\theta_k - \theta(t_k)\|_2 \leq e^{t_k}\|g_k\|_2$, we obtain that

$$\|\theta_k\|_2 \leq \|\theta(t_k)\|_2 + \|\theta(t_k) - \theta_k\|_2 \leq \|\theta(t_k)\|_2 + e^{t_k}\|g_k\|_2 \leq \|\theta(t_k)\|_2 + e^{t_k}\frac{C_0(e^{\alpha_k} - 1)}{(e^{t_k} - 1)}\|\theta_k\|_2 \,,$$

which implies that

$$\|\theta_k\|_2 \leq \frac{\|\theta(t_k)\|_2}{1 - \frac{C_0(e^{\alpha_k}-1)}{(1-e^{-t_k})}} \,. \tag{S8}$$

Similarly, we have

$$\|\theta(t_k)\|_2 \leq \|\theta_k\|_2 + \|\theta(t_k) - \theta_k\|_2 \leq \|\theta_k\|_2 + e^{t_k}\|g_k\|_2 \leq \|\theta_k\|_2 + \frac{C_0(e^{\alpha_k} - 1)}{(1 - e^{-t_k})}\|\theta_k\|_2 \,,$$

which implies that

$$\frac{\|\theta_k\|_2}{\|\theta(t_k)\|_2} \geq \frac{1}{1 + \frac{C_0(e^{\alpha_k}-1)}{(1-e^{-t_k})}} \,.$$

Combining this with (S8), it proves (S7). This completes the proof of Lemma S2. ∎

We next present a supporting lemma to be used in the proof of Theorem 4.

**Lemma S3.** *For any* $\epsilon \in (0, 1]$*, let* $(e^{\alpha_1} - 1)^2 = c_1^2\epsilon$ *and* $\alpha_{k+1} = \ln(1 + c_2 e^{t_k/2}(e^{\alpha_1} - 1))$*, where* $c_1, c_2 > 0$ *are constants and* $t_k = \sum_{i=1}^k \alpha_i$*. Define* $k^\star := \max\{k : t_k \leq \ln(\epsilon^{-1})\}$*. Then,*

$$k^\star < \frac{1 + c_1 c_2 + \sqrt{1 + c_1 c_2}}{c_1 c_2\sqrt{\epsilon}\sqrt{1 + c_1\sqrt{\epsilon}}} \,. \tag{S9}$$

*Proof of Lemma S3.* It is easy to see that $\alpha_k$ is strictly increasing for $k \geq 2$ and by definition of $k^\star$, we have that

$$\alpha_{k+1} \leq \ln(1 + c_1 c_2) \text{ for any } 1 \leq k \leq k^\star, \tag{S10}$$

which implies that $e^{\alpha_{k+1}/2} \leq (1 + c_1 c_2)^{1/2}$ for any $1 \leq k \leq k^\star$. Moreover, since $e^{\alpha_{k+1}} = 1 + c_1 c_2\sqrt{\epsilon}e^{t_k/2}$, we have that

$$e^{-t_k/2} - e^{-t_{k+1}/2} = \frac{e^{\alpha_{k+1}/2} - 1}{e^{t_{k+1}/2}} = \frac{e^{-t_{k+1}/2}(e^{\alpha_{k+1}} - 1)}{e^{\alpha_{k+1}/2} + 1} = \frac{e^{-\alpha_{k+1}/2}c_1 c_2\sqrt{\epsilon}}{e^{\alpha_{k+1}/2} + 1} = \frac{c_1 c_2\sqrt{\epsilon}}{e^{\alpha_{k+1}} + e^{\alpha_{k+1}/2}}$$

for any $k \geq 1$. Therefore, for any $1 \leq k \leq k^\star$,

$$e^{-t_1/2} - e^{-t_{k^\star+1}/2} = \sum_{i=1}^{k^\star}\left(e^{-t_i/2} - e^{-t_{i+1}/2}\right) = \sum_{i=1}^{k^\star}\frac{c_1 c_2\sqrt{\epsilon}}{e^{\alpha_{i+1}} + e^{\alpha_{i+1}/2}} \geq \frac{k^\star c_1 c_2\sqrt{\epsilon}}{1 + c_1 c_2 + \sqrt{1 + c_1 c_2}} \,,$$

which implies (S9). This completes the proof of Lemma S3. ∎

# 4 Some properties of $\ell_2$-regularized solution path

We present some properties of the solution path to be used later in other proofs.

**Corollary S1.** *Suppose that* $L_n(\theta)$ *is a closed proper convex function. Let* $\theta(t)$ *denote the* $\ell_2$*-regularized solution path defined by* (2)*. Then*

 *(i)* $\|\theta(t)\|_2$ *is nondecreasing in* $t$ *and* $L_n(\theta(t))$ *is nonincreasing in* $t$*;*

*(ii)* $\|\theta(t)\|_2/(e^t - 1)$ *is nonincreasing in $t$;*

*(iii)* *if $L_n(\theta)$ is a continuous and the minimum $\ell_2$ norm minimizer of $L_n(\theta)$, denoted as $\theta^\star$, is finite, then $\lim_{t\to\infty} \theta(t) = \theta^\star$.*

This corollary shows that the $\ell_2$ norm of the solutions along the path is monotone, and the solution $\theta(t)$ converges to the minimum $\ell_2$ norm minimizer of $L_n(\theta)$ as $t$ goes to infinity if it is finite. We also remark that part (iii) has already been established in Theorem 8 of Suggala et al. [2018]. We include these properties here to make the article largely self-contained.

*Proof of Corollary S1.* To prove (i), rearranging terms in (S5), we obtain that

$$(C(t') - C(t)) \left(\|\theta(t')\|_2^2 - \|\theta(t)\|_2^2\right) \geq (C(t) + C(t'))\|\theta(t) - \theta(t')\|_2^2 \geq 0\,,$$

which implies that $\|\theta(t)\|_2$ is nondecreasing in $t$. For nonincreasingness of $L_n(\theta(t))$, note that

$$
\begin{aligned}
C(t')L_n(\theta(t')) + \frac{1}{2}\|\theta(t')\|_2^2 &\leq C(t')L_n(\theta(t)) + \frac{1}{2}\|\theta(t)\|_2^2 \\
&\leq (C(t') - C(t))L_n(\theta(t)) + C(t)L_n(\theta(t')) + \frac{1}{2}\|\theta(t')\|_2^2\,,
\end{aligned}
$$

which implies that $(C(t') - C(t))(L_n(\theta(t')) - L_n(\theta(t))) \leq 0$. Hence, if $C(t') - C(t) > 0$ then $L_n(\theta(t')) \leq L_n(\theta(t))$, which proves that $L_n(\theta(t))$ is nonincreasing in $t$.

To prove (ii), using (S3), we have that for any $t > t'$ and $\theta(t) \neq \theta(t')$,

$$\|\theta(t)\|_2 - \|\theta(t')\|_2 \leq \|\theta(t') - \theta(t)\|_2 \leq (C(t) - C(t'))\|\theta(t)\|_2/C(t)\,, \tag{S11}$$

which implies that

$$\|\theta(t)\|_2/C(t) \leq \|\theta(t')\|_2/C(t')\,. \tag{S12}$$

This also holds when $\theta(t) = \theta(t')$ because $C(t)$ is an increasing function. This proves part (ii).

Lastly, we prove part (iii). Denote by $\theta^\star$ the minimum $\ell_2$ norm minimizer of $L_n(\theta)$. Next, we show that $\theta(t)$ converges to $\theta^\star$ as $t \to \infty$ if $\theta^\star$ is finite. Note that $\mathbf{0} \in \partial L_n(\theta^\star)$ and $\mathbf{0} \in C(t)\partial L_n(\theta(t)) + \theta(t)$. As a result,

$$\mathbf{0} \in C(t)\left(\partial L_n(\theta(t)) - \partial L_n(\theta^\star)\right) + \theta(t)\,,$$

where $A - B$ denotes the set $\{a - b : a \in A \text{ and } b \in B\}$. Multiplying $\theta(t) - \theta^\star$ on both sides, we obtain that

$$(\theta(t) - \theta^\star)^\top \theta(t) \in -C(t)(\theta(t) - \theta^\star)^\top \left(\partial L_n(\theta(t)) - \partial L_n(\theta^\star)\right)\,,$$

which implies that $(\theta(t) - \theta^\star)^\top \theta(t) \leq 0$. Therefore, $\|\theta(t)\|_2^2 \leq (\theta^\star)^\top \theta(t) \leq \|\theta^\star\|_2\|\theta(t)\|_2$, which implies that $\|\theta(t)\|_2 \leq \|\theta^\star\|_2 < \infty$ for any $t \geq 0$. Denote by $\bar{\theta}$ the limit of any converging subsequence $\theta(t_k)$, that is, $\bar{\theta} = \lim_{k\to\infty} \theta(t_k)$ for some $t_k \to \infty$. Then, $\|\bar{\theta}\|_2 = \lim_{k\to\infty} \|\theta(t_k)\|_2 \leq \|\theta^\star\|_2$. Next, we show that $\bar{\theta}$ must also be a minimizer of $L_n(\theta)$. To this end, note that $L_n(\bar{\theta}) = \lim_{k\to\infty} L_n(\theta(t_k))$ by using the continuity of $L_n(\theta)$ in $\theta$. Moreover, by optimality of $\theta(t_k)$,

$$L_n(\theta(t_k)) \leq L_n(\theta(t_k)) + \frac{1}{2C(t_k)}\|\theta(t_k)\|_2^2 \leq L_n(\theta^\star) + \frac{1}{2C(t_k)}\|\theta^\star\|_2^2\,. \tag{S13}$$

By letting $k \to \infty$ and using the fact that $L_n(\bar{\theta}) = \lim_{k\to\infty} L_n(\theta(t_k))$ due to continuity of $L_n(\theta)$, we have that

$$L_n(\bar{\theta}) = \lim_{k\to\infty} L_n(\theta(t_k)) \leq \lim_{k\to\infty} \left(L_n(\theta^\star) + \frac{1}{2C(t_k)}\|\theta^\star\|_2^2\right) = L_n(\theta^\star)\,,$$

where the last step uses the assumption that $\|\theta^\star\|_2 < \infty$. This proves that $\bar{\theta}$ must also be a minimizer of $L_n(\theta)$.

Now if $\bar{\theta} \neq \theta^\star$, then their convex combination $\frac{1}{2}(\bar{\theta} + \theta^\star)$ must also be a minimizer of $L_n(\theta)$ due to the convexity of $L_n(\theta)$. On the other hand, the convex combination has strictly smaller norm than that of $\theta^\star$, because $\|\frac{1}{2}(\bar{\theta} + \theta^\star)\|_2 < \frac{1}{2}(\|\bar{\theta}\| + \|\theta^\star\|_2) \leq \|\theta^\star\|_2$. This contradicts with the definition of $\theta^\star$. Hence, we must have $\lim_{k\to\infty} \theta(t_k) = \bar{\theta} = \theta^\star$ for every converging subsequence $\theta(t_k)$. Consequently, the sequence $\theta(t)$ must converge to $\theta^\star$. This completes the proof of Corollary S1. ∎

# 5 Proof of Theorem 1

*Proof of Theorem 1.* For any $t \in [t_k, t_{k+1}]$, we let $w_k = \frac{t_{k+1}-t}{t_{k+1}-t_k}$, for $k = 0, 1, \ldots, N-1$. Then $\tilde{\theta}(t) = w_k \theta_k + (1-w_k)\theta_{k+1}$. By convexity of $f_t(\cdot)$, we have $f_t(\tilde{\theta}(t)) \leq w_k f_t(\theta_k) + (1-w_k)f_t(\theta_{k+1})$. Thus,

$$f_t(\tilde{\theta}(t)) - f_t(\theta(t)) \leq w_k(f_t(\theta_k) - f_t(\theta(t))) + (1-w_k)(f_t(\theta_{k+1}) - f_t(\theta(t))). \tag{S14}$$

For any $k = 1, \ldots, N-1$, the term $f_t(\theta_k) - f_t(\theta(t))$ in (S14) can be bounded as follows:

$$f_t(\theta_k) - f_t(\theta(t)) \leq \frac{1}{2e^{-t}} \|\nabla f_t(\theta_k)\|_2^2 = \frac{e^t}{2} \left\| \frac{1-e^{-t}}{1-e^{-t_k}}\nabla f_{t_k}(\theta_k) + \frac{e^{-t}-e^{-t_k}}{1-e^{-t_k}}\theta_k \right\|_2^2$$

$$\leq e^t \left(\frac{1-e^{-t}}{1-e^{-t_k}}\right)^2 \|\nabla f_{t_k}(\theta_k)\|_2^2 + e^t \left(\frac{e^{-t}-e^{-t_k}}{1-e^{-t_k}}\right)^2 \|\theta_k\|_2^2$$

$$= e^t \left(\frac{1-e^{-t}}{1-e^{-t_k}}\right)^2 \|g_k\|_2^2 + e^t \left(\frac{e^{-t}-e^{-t_k}}{1-e^{-t_k}}\right)^2 \|\theta_k\|_2^2,$$

where the first inequality uses the fact that $f_t(\cdot)$ is $e^{-t}$-strongly convex and (S2). Similarly, we can bound the term $f_t(\theta_{k+1}) - f_t(\theta(t))$ by

$$e^t \left(\frac{1-e^{-t}}{1-e^{-t_{k+1}}}\right)^2 \|g_{k+1}\|_2^2 + e^t \left(\frac{e^{-t}-e^{-t_{k+1}}}{1-e^{-t_{k+1}}}\right)^2 \|\theta_{k+1}\|_2^2$$

for any $k = 0, 1, \ldots, N-1$. Combining these two bounds, we have that

$$w_k(f_t(\theta_k) - f_t(\theta(t))) + (1-w_k)(f_t(\theta_{k+1}) - f_t(\theta(t)))$$

$$\leq e^{t_{k+1}} \max \left\{ \left(\frac{1-e^{-t_{k+1}}}{1-e^{-t_k}}\right)^2 \|g_k\|_2^2, \|g_{k+1}\|_2^2 \right\} +$$

$$(e^{-t_k} - e^{-t_{k+1}})^2 \max \left\{ \frac{e^{t_{k+1}}\|\theta_k\|_2^2}{(1-e^{-t_k})^2}, \frac{e^{t_k}\|\theta_{k+1}\|_2^2}{(1-e^{-t_{k+1}})^2} \right\},$$

for any $k = 1, \ldots, N-1$. This proves (5).

When $k = 0$, the term $f_t(\theta_k) - f_t(\theta(t))$ in (S14) can be bounded as follows

$$f_t(\theta_0) - f_t(\theta(t)) = f_t(\mathbf{0}) - f_t(\theta(t)) \leq \frac{1}{2e^{-t}}\|\nabla f_t(\mathbf{0})\|_2^2 = \frac{e^t(1-e^{-t})^2}{2}\|\nabla L_n(\mathbf{0})\|_2^2$$

for any $0 \leq t < t_1$, where we have used (S2) in the above inequality. Following a similar argument as before, we obtain that

$$w_0(f_t(\theta_0)-f_t(\theta(t)))+(1-w_0)(f_t(\theta_1)-f_t(\theta(t))) \leq \frac{e^{t_1}(1-e^{-t_1})^2}{2}\|\nabla L_n(\mathbf{0})\|_2^2+\max\left(e^{t_1}\|g_1\|_2^2,\ \|\theta_1\|_2^2\right)$$

for any $t \in [0, t_1]$. This proves (4).

Now we bound $f_t(\tilde{\theta}(t)) - f_t(\theta(t))$ when $t_N < t \leq t_{\max}$. Toward this end, notice that

$$f_t(\tilde{\theta}(t)) - f_t(\theta(t)) = f_t(\theta_N) - f_t(\theta(t))$$
$$= \frac{1-e^{-t}}{1-e^{-t_N}}(f_{t_N}(\theta_N) - f_{t_N}(\theta(t_N))) + \frac{e^{-t_N}-e^{-t}}{2(1-e^{-t_N})}(\|\theta(t_N)\|_2^2 - \|\theta_N\|_2^2) + f_t(\theta(t_N)) - f_t(\theta(t)).$$

Next, we bound these three terms separately. For the first term, by using (S2), we have that $f_{t_N}(\theta_N) - f_{t_N}(\theta(t_N)) \leq 2^{-1}e^{t_N}\|g_N\|_2^2$. Using this, we obtain that

$$\frac{1-e^{-t}}{1-e^{-t_N}}(f_{t_N}(\theta_N) - f_{t_N}(\theta(t_N))) \leq \frac{(1-e^{-t})e^{t_N}}{2(1-e^{-t_N})}\|g_N\|_2^2. \tag{S15}$$

We next bound the third term. By optimality of $\theta(t_N)$, we have $f_{t_N}(\theta(t_N)) \leq f_{t_N}(\theta(t))$, which in turn implies that $L_n(\theta(t_N)) - L_n(\theta(t)) \leq .5(e^{t_N} - 1)^{-1}\left(\|\theta(t)\|_2^2 - \|\theta(t_N)\|_2^2\right)$. Using this, the

third term can be bounded as follows

$$f_t(\theta(t_N)) - f_t(\theta(t)) = (1 - e^{-t})(L_n(\theta(t_N)) - L_n(\theta(t))) + \frac{e^{-t}}{2}\left(\|\theta(t_N)\|_2^2 - \|\theta(t)\|_2^2\right)$$

$$\leq \frac{1 - e^{-t}}{2(e^{t_N} - 1)}\left(\|\theta(t)\|_2^2 - \|\theta(t_N)\|_2^2\right) + \frac{e^{-t}}{2}\left(\|\theta(t_N)\|_2^2 - \|\theta(t)\|_2^2\right)$$

$$= \frac{e^{-t_N} - e^{-t}}{2(1 - e^{-t_N})}\left(\|\theta(t)\|_2^2 - \|\theta(t_N)\|_2^2\right).$$

Combining the bounds for the first term and the third term, we obtain that

$$\sup_{t_N < t \leq t_{\max}} \{f_t(\tilde{\theta}(t)) - f_t(\theta(t))\}$$

$$\leq \sup_{t_N < t \leq t_{\max}} \left\{\frac{(1 - e^{-t})e^{t_N}}{2(1 - e^{-t_N})}\|g_N\|_2^2 + \frac{e^{-t_N} - e^{-t}}{2(1 - e^{-t_N})}(\|\theta(t)\|_2^2 - \|\theta_N\|_2^2)\right\} \tag{S16}$$

$$\leq \frac{e^{t_N}}{2(1 - e^{-t_N})}\|g_N\|_2^2 + \frac{1}{2(e^{t_N} - 1)}\|\theta(t_{\max})\|_2^2,$$

which implies (6). This completes the proof of Theorem 1. $\blacksquare$

# 6 Proof of Theorem 2

*Proof of Theorem 2.* Using (8), we obtain that

$$e^{t_{k+1}}\|g_{k+1}\|_2^2 \leq e^{t_{k+1}}\left(C_0\frac{(e^{\alpha_{k+1}} - 1)}{(e^{t_{k+1}} - 1)}\|\theta_{k+1}\|_2\right)^2 \leq C_0^2 e^{\alpha_{\max}}(e^{-t_k} - e^{-t_{k+1}})^2\frac{e^{t_k}\|\theta_{k+1}\|_2^2}{(1 - e^{-t_{k+1}})^2},$$

and

$$e^{t_{k+1}}\left(\frac{1 - e^{-t_{k+1}}}{1 - e^{-t_k}}\right)^2\|g_k\|_2^2 \leq e^{t_{k+1}}\left(\frac{1 - e^{-t_{k+1}}}{1 - e^{-t_k}}\right)^2\left(C_0\frac{(e^{\alpha_k} - 1)}{(e^{t_k} - 1)}\|\theta_k\|_2\right)^2,$$

$$\leq C_0^2(e^{\alpha_{\max}} + e^{\alpha_{\max}/2} + 1)^2(e^{-t_k} - e^{-t_{k+1}})^2\frac{e^{t_{k+1}}\|\theta_k\|_2^2}{(1 - e^{-t_k})^2}$$

where we have used the fact that $\alpha_{k+1} \geq \alpha_k/2$ and

$$\frac{e^{\alpha_k} - 1}{1 - e^{-t_k}}\frac{e^{\alpha_{k+1}} - e^{-t_k}}{e^{\alpha_{k+1}} - 1} \leq \frac{e^{\alpha_k} - 1}{1 - e^{-t_k}}\left(1 + \frac{1 - e^{-t_k}}{e^{\alpha_{k+1}} - 1}\right) \leq \frac{e^{\alpha_k} - 1}{1 - e^{-t_k}}\left(1 + \frac{1 - e^{-t_k}}{e^{\alpha_k/2} - 1}\right)$$

$$= \frac{e^{\alpha_k} - 1}{1 - e^{-t_k}} + e^{\alpha_k/2} + 1 \leq e^{\alpha_k} + e^{\alpha_k/2} + 1 \leq e^{\alpha_{\max}} + e^{\alpha_{\max}/2} + 1.$$

Combining, we get

$$e^{t_{k+1}}\max\left\{\left(\frac{1 - e^{-t_{k+1}}}{1 - e^{-t_k}}\right)^2\|g_k\|_2^2, \|g_{k+1}\|_2^2\right\}$$

$$\leq C_0^2(e^{\alpha_{\max}} + e^{\alpha_{\max}/2} + 1)^2(e^{-t_k} - e^{-t_{k+1}})^2\max\left\{\frac{e^{t_{k+1}}\|\theta_k\|_2^2}{(1 - e^{-t_k})^2}, \frac{e^{t_k}\|\theta_{k+1}\|_2^2}{(1 - e^{-t_{k+1}})^2}\right\},$$

which together with (5), we obtain that

$$\sup_{t \in [t_k, t_{k+1}]} \{f_t(\tilde{\theta}(t)) - f_t(\theta(t))\}$$

$$\leq (1 + C_1^2)(e^{-t_k} - e^{-t_{k+1}})^2\max\left\{\frac{e^{t_{k+1}}\|\theta_k\|_2^2}{(1 - e^{-t_k})^2}, \frac{e^{t_k}\|\theta_{k+1}\|_2^2}{(1 - e^{-t_{k+1}})^2}\right\}, \tag{S17}$$

where $C_1 = C_0(e^{\alpha_{\max}} + e^{\alpha_{\max}/2} + 1)$. Therefore,

$$\max_{1 \leq k \leq N-1} \sup_{t \in [t_k, t_{k+1}]} \left\{ f_t(\tilde{\theta}(t)) - f_t(\theta(t)) \right\}$$

$$\leq \max_{1 \leq k \leq N-1} (1 + C_1^2)(e^{-t_k} - e^{-t_{k+1}})^2 \max \left\{ \frac{e^{t_{k+1}} \|\theta_k\|_2^2}{(1 - e^{-t_k})^2}, \frac{e^{t_k} \|\theta_{k+1}\|_2^2}{(1 - e^{-t_{k+1}})^2} \right\}$$

$$\leq (1 + C_1^2) \max_{1 \leq k \leq N} \left\{ \frac{e^{-t_k} \|\theta_k\|_2^2}{(1 - e^{-t_k})^2} \max \left( e^{-\alpha_{k+1}} (e^{\alpha_{k+1}} - 1)^2, e^{-\alpha_k} (e^{\alpha_k} - 1)^2 \right) \right\}$$

$$\leq 4(1 + C_1^2) \max_{1 \leq k \leq N} \left\{ \frac{e^{-t_k} \|\theta_k\|_2^2 (e^{\alpha_{k+1}} - 1)^2}{(1 - e^{-t_k})^2} \right\} .$$

Using (4) and (8), we have that

$$\sup_{t \in [0, t_1]} \left\{ f_t(\tilde{\theta}(t)) - f_t(\theta(t)) \right\} \leq (3/2) \cdot \max \left( (e^{\alpha_1} - 1)^2 \|\nabla L_n(\mathbf{0})\|_2^2, \|\theta_1\|_2^2 \right) \tag{S18}$$

Therefore, if $t_{N-1} \leq t_{\max} \leq t_N$ for some $N \geq 1$, we have

$$\sup_{0 \leq t \leq t_{\max}} \left\{ f_t(\tilde{\theta}(t)) - f_t(\theta(t)) \right\} \leq \sup_{0 \leq t \leq t_N} \left\{ f_t(\tilde{\theta}(t)) - f_t(\theta(t)) \right\}$$

$$\leq 4(1 + C_1^2) \max \left\{ (e^{\alpha_1} - 1)^2 \|\nabla L_n(\mathbf{0})\|_2^2, \max_{1 \leq k \leq N} \left( \frac{e^{-t_k} \|\theta_k\|_2^2 (e^{\alpha_{k+1}} - 1)^2}{(1 - e^{-t_k})^2} \right) \right\},$$

which implies (10).

Now if $t_N < t_{\max}$, by using (S16), we have that $\sup_{t_N < t \leq t_{\max}} \left\{ f_t(\tilde{\theta}(t)) - f_t(\theta(t)) \right\}$ can be upper bounded by

$$\sup_{t_N < t \leq t_{\max}} \left\{ \frac{(1 - e^{-t}) e^{t_N}}{2(1 - e^{-t_N})} \|g_N\|_2^2 + \frac{e^{-t_N} - e^{-t}}{2(1 - e^{-t_N})} (\|\theta(t)\|_2^2 - \|\theta_N\|_2^2) \right\}$$

$$\leq \sup_{t_N < t \leq t_{\max}} \left\{ \frac{(1 - e^{-t}) e^{t_N}}{2(1 - e^{-t_N})} \left( C_0 \frac{(e^{\alpha_N} - 1)}{(e^{t_N} - 1)} \|\theta_N\|_2 \right)^2 + \frac{e^{-t_N} - e^{-t}}{2(1 - e^{-t_N})} (\|\theta(t)\|_2^2 - \|\theta_N\|_2^2) \right\}$$

$$\leq \frac{e^{-t_N}}{2} \left( C_0 \frac{(e^{\alpha_N} - 1)}{(1 - e^{-t_N})} \right)^2 \|\theta_N\|_2^2 + \frac{1 - e^{-(t_{\max} - t)}}{2(e^{t_N} - 1)} \|\theta(t_{\max})\|_2^2$$

$$\leq \frac{1 - e^{-\max(\alpha_N, t_{\max} - t_N)}}{e^{t_N} - 1} \max \left\{ \|\theta_N\|_2^2, \|\theta(t_{\max})\|_2^2 \right\},$$

provided that $C_0 e^{\alpha_{\max}} < 1/2$, or sufficiently $C_0 \leq 1/4$. Combining this with the previous bounds, it proves (9). This completes the proof of Theorem 2. ∎

# 7   Approximation-error bound for $\|\tilde{\theta}(t) - \theta(t)\|_2$

**Corollary S2.** *Under the assumptions in Theorem 2, we have that*

$$\sup_{0 \leq t \leq t_{\max}} \|\tilde{\theta}(t) - \theta(t)\|_2 \leq 5 \max \left\{ (e^{\alpha_1} - 1) \|\nabla L_n(\mathbf{0})\|_2, \max_{1 \leq k \leq N} \left( \frac{\|\theta_k\|_2 (e^{\alpha_{k+1}} - 1)}{1 - e^{-t_k}} \right) \right\} \tag{S19}$$

*when $t_{N-1} \leq t_{\max} \leq t_N$ for some $N \geq 1$, and*

$$\sup_{t_N < t \leq t_{\max}} \|\tilde{\theta}(t) - \theta(t)\|_2 \leq \frac{(e^{\alpha_N} - 1) \|\theta_N\|_2}{4(1 - e^{-t_N})} + 2 \sup_{t_N \leq t \leq t_{\max}} \|\theta(t) - \theta(t_{\max})\|_2 \tag{S20}$$

*when $t_N < t_{\max}$.*

*Proof of Corollary S2.* By using (S17) and (S2), we obtain that

$$\sup_{t \in [t_k, t_{k+1}]} \|\tilde{\theta}(t) - \theta(t)\|_2^2 \leq 2 \sup_{t \in [t_k, t_{k+1}]} e^t \left( f_t(\tilde{\theta}(t)) - f_t(\theta(t)) \right)$$

$$\leq 2(1 + C_1^2) e^{t_{k+1}} (e^{-t_k} - e^{-t_{k+1}})^2 \max \left\{ \frac{e^{t_{k+1}} \|\theta_k\|_2^2}{(1 - e^{-t_k})^2}, \frac{e^{t_k} \|\theta_{k+1}\|_2^2}{(1 - e^{-t_{k+1}})^2} \right\} \tag{S21}$$

where $C_1 = C_0(e^{\alpha_{\max}} + e^{\alpha_{\max}/2} + 1)$. Therefore,

$$\max_{1\leq k\leq N-1} \sup_{t\in[t_k,t_{k+1}]} \|\tilde{\theta}(t) - \theta(t)\|_2^2$$

$$\leq 2(1+C_1^2) \max_{1\leq k\leq N-1} e^{t_{k+1}}(e^{-t_k} - e^{-t_{k+1}})^2 \max\left\{ \frac{e^{t_{k+1}}\|\theta_k\|_2^2}{(1-e^{-t_k})^2}, \frac{e^{t_k}\|\theta_{k+1}\|_2^2}{(1-e^{-t_{k+1}})^2} \right\}$$

$$\leq 2(1+C_1^2) \max_{1\leq k\leq N} \left\{ \frac{\|\theta_k\|_2^2}{(1-e^{-t_k})^2} \max\left((e^{\alpha_{k+1}}-1)^2, e^{-\alpha_k}(e^{\alpha_k}-1)^2\right) \right\}$$

$$\leq 8(1+C_1^2) \max_{1\leq k\leq N} \left\{ \frac{\|\theta_k\|_2^2(e^{\alpha_{k+1}}-1)^2}{(1-e^{-t_k})^2} \right\}.$$

Moreover, using (4) and (S2), we have that

$$\sup_{t\in[0,t_1]} \|\tilde{\theta}(t) - \theta(t)\|_2^2 \leq 2 \sup_{t\in[0,t_1]} e^t \left( f_t(\tilde{\theta}(t)) - f_t(\theta(t)) \right)$$

$$\leq 3e^{\alpha_1} \max\left( (e^{\alpha_1}-1)^2\|\nabla L_n(\mathbf{0})\|_2^2, \|\theta_1\|_2^2 \right) \tag{S22}$$

Therefore, if $t_{N-1} \leq t_{\max} \leq t_N$ for some $N \geq 1$, we have

$$\sup_{0\leq t\leq t_{\max}} \|\tilde{\theta}(t) - \theta(t)\|_2^2 \leq \sup_{0\leq t\leq t_N} \|\tilde{\theta}(t) - \theta(t)\|_2^2$$

$$\leq 8(1+C_1^2)\max\left\{ (e^{\alpha_1}-1)^2\|\nabla L_n(\mathbf{0})\|_2^2, \max_{1\leq k\leq N}\left( \frac{\|\theta_k\|_2^2(e^{\alpha_{k+1}}-1)^2}{(1-e^{-t_k})^2} \right) \right\},$$

which implies (S19), because $8(1+C_1^2) = 8\left(1 + C_0^2(e^{\alpha_{\max}} + e^{\alpha_{\max}/2} + 1)^2\right) \leq 25$.

To bound $\sup_{t_N < t\leq t_{\max}} \|\tilde{\theta}(t) - \theta(t)\|_2$, note that for any $t_N < t \leq t_{\max}$, we have that

$$\|\tilde{\theta}(t) - \theta(t)\|_2 = \|\theta_N - \theta(t)\|_2 \leq \|\theta_N - \theta(t_N)\|_2 + \|\theta(t_N) - \theta(t)\|_2$$

$$\leq e^{t_N}\|g_N\|_2 + \|\theta(t_N) - \theta(t)\|_2$$

$$\leq C_0\frac{(e^{\alpha_N}-1)\|\theta_N\|_2}{1-e^{-t_N}} + \|\theta(t_N) - \theta(t_{\max})\|_2 + \|\theta(t) - \theta(t_{\max})\|_2.$$

Hence,

$$\sup_{t_N < t\leq t_{\max}} \|\tilde{\theta}(t) - \theta(t)\|_2 \leq C_0\frac{(e^{\alpha_N}-1)\|\theta_N\|_2}{1-e^{-t_N}} + 2 \sup_{t_N \leq t\leq t_{\max}} \|\theta(t) - \theta(t_{\max})\|_2,$$

which proves (S20). This completes the proof of Corollary S2. ∎

## 8   Proof of Theorem 3

*Proof of Theorem 3.* We first show that $\alpha_{k+1} \geq \alpha_k/2$. If $\alpha_{k+1} = \alpha_{\max}$ or $2\alpha_k$, then trivially $\alpha_{k+1} \geq \alpha_k/2$. Now we assume that $\alpha_{k+1} = A_k$, where

$$A_k := \ln\left( 1 + \frac{c_1(e^{\alpha_1}-1)\|\nabla L_n(\mathbf{0})\|_2 e^{t_k/2}(1-e^{-t_k})}{\|\theta_k\|_2} \right).$$

When $k = 1$, using (S7) and the fact that $e^{\alpha_1} - 1 \leq \sqrt{\epsilon}/\|\nabla L_n(\mathbf{0})\|_2$, we obtain that

$$\frac{e^{\alpha_2}-1}{e^{\alpha_1/2}-1} = (e^{\alpha_1/2}+1)\frac{e^{\alpha_2}-1}{e^{\alpha_1}-1} \geq (e^{\alpha_1/2}+1)\frac{e^{A_1}-1}{e^{\alpha_1}-1} = (e^{\alpha_1/2}+1)e^{\alpha_1/2}\frac{c_1\sqrt{\epsilon}(1-e^{-t_1})}{\|\theta_1\|_2(e^{\alpha_1}-1)}$$

$$\geq (e^{\alpha_1/2}+1)e^{\alpha_1/2}\frac{c_1\sqrt{\epsilon}(1-e^{-t_1})}{\|\theta(t_1)\|_2(e^{\alpha_1}-1)}\left(1 - \frac{C_0(e^{\alpha_1}-1)}{(1-e^{-t_1})}\right)$$

$$\geq (e^{-\alpha_1/2}+1)\frac{c_1\sqrt{\epsilon}}{\|\nabla L_n(\mathbf{0})\|_2(e^{\alpha_1}-1)}(1-C_0 e^{\alpha_1}) \geq (e^{-\alpha_1/2}+1)(1-C_0 e^{\alpha_1}) \geq 1,$$

provided that $c_1 \geq 1$ and $C_0 \leq 4^{-1}$. This implies that $\alpha_2 \geq \alpha_1/2$.

When $k \geq 2$, note that $\alpha_k \leq A_{k-1}$ and

$$
\begin{aligned}
\frac{e^{\alpha_{k+1}} - 1}{e^{\alpha_k/2} - 1} &= (e^{\alpha_k/2} + 1)\frac{e^{\alpha_{k+1}} - 1}{e^{\alpha_k} - 1} \geq (e^{\alpha_k/2} + 1)\frac{e^{A_k} - 1}{e^{A_{k-1}} - 1} \\
&= (e^{\alpha_k/2} + 1)e^{\alpha_k/2}\frac{(e^{\alpha_1} - 1)\|\nabla L_n(\mathbf{0})\|_2(1 - e^{-t_k})}{\|\theta_k\|_2} \frac{\|\theta_{k-1}\|_2}{(e^{\alpha_1} - 1)\|\nabla L_n(\mathbf{0})\|_2(1 - e^{-t_{k-1}})} \\
&\geq (e^{-\alpha_k/2} + 1)\frac{(e^{t_k} - 1)\|\theta_{k-1}\|_2}{(e^{t_{k-1}} - 1)\|\theta_k\|_2}.
\end{aligned}
\tag{S23}
$$

Now using (S7) with $C_0 \leq 4^{-1}$, we have that

$$
\frac{(e^{t_k} - 1)\|\theta_{k-1}\|_2}{(e^{t_{k-1}} - 1)\|\theta_k\|_2} \geq \frac{(e^{t_k} - 1)\|\theta(t_{k-1})\|_2}{(e^{t_{k-1}} - 1)\|\theta(t_k)\|_2}\frac{1 - \frac{(e^{\alpha_k} - 1)}{4(1 - e^{-t_k})}}{1 + \frac{(e^{\alpha_k} - 1)}{4(1 - e^{-t_k})}} \geq \frac{1 - \frac{(e^{\alpha_k} - 1)}{4(1 - e^{-t_k})}}{1 + \frac{(e^{\alpha_k} - 1)}{4(1 - e^{-t_k})}}
\tag{S24}
$$

where the last inequality uses part (ii) of Corollary S1. Combining this with (S23), we obtain that

$$
\frac{e^{\alpha_{k+1}} - 1}{e^{\alpha_k/2} - 1} \geq (e^{-\alpha_k/2} + 1)\frac{1 - \frac{(e^{\alpha_k} - 1)}{4(1 - e^{-t_k})}}{1 + \frac{(e^{\alpha_k} - 1)}{4(1 - e^{-t_k})}}.
\tag{S25}
$$

Therefore, to prove $\alpha_{k+1} \geq \alpha_k/2$, it suffices to show that the RHS of the above inequality is no smaller than 1. To this end, using the fact that $\alpha_k \leq \alpha_{\max} \leq 1/5$ for all $k \geq 1$, we have

$$
\begin{aligned}
\frac{(e^{\alpha_k} - 1)}{(1 - e^{-t_k})} &= \frac{e^{\alpha_k}(e^{\alpha_k} - 1)}{e^{\alpha_k} - e^{-t_{k-1}}} = \frac{e^{\alpha_k}}{1 + \frac{1 - e^{-t_{k-1}}}{e^{\alpha_k} - 1}} \leq \frac{e^{2\alpha_{k-1}}}{1 + \frac{1 - e^{-t_{k-1}}}{e^{2\alpha_{k-1}} - 1}} \leq \frac{e^{2\alpha_{k-1}}}{1 + \frac{1 - e^{-\alpha_{k-1}}}{e^{2\alpha_{k-1}} - 1}} \\
&\leq \frac{e^{3\alpha_{k-1}}(e^{\alpha_{k-1}} + 1)}{e^{2\alpha_{k-1}} + e^{\alpha_{k-1}} + 1} = e^{2\alpha_{k-1}}\left(1 - \frac{1}{e^{2\alpha_{k-1}} + e^{\alpha_{k-1}} + 1}\right) \\
&\leq e^{2/5}\left(1 - \frac{1}{e^{2/5} + e^{1/5} + 1}\right) \leq 1.09.
\end{aligned}
$$

for any $k \geq 2$, and when $k = 1$, we have

$$
\frac{(e^{\alpha_k} - 1)}{(1 - e^{-t_k})} = e^{\alpha_1} \leq \exp(1/5) < 1.22.
\tag{S26}
$$

Combining this with (S25), we have that

$$
\frac{e^{\alpha_{k+1}} - 1}{e^{\alpha_k/2} - 1} \geq (e^{-\alpha_k/2} + 1)\frac{1 - \frac{(e^{\alpha_k} - 1)}{4(1 - e^{-t_k})}}{1 + \frac{(e^{\alpha_k} - 1)}{4(1 - e^{-t_k})}} \geq (e^{-1/10} + 1)\frac{1 - 1.22/4}{1 + 1.22/4} > 1,
$$

which proves that $\alpha_{k+1} \geq \alpha_k/2$. Hence, $\alpha_k$ satisfies all the conditions in Theorem 2.

Next, we show that the algorithm will terminate in finite steps. We first show that $t_k$ diverges. To this end, using (S6) and (S7), we have

$$
\begin{aligned}
\frac{\epsilon^{1/2}e^{t_k/2}(1 - e^{-t_k})}{\|\theta_k\|_2} &\geq \frac{e^{t_k/2}(e^{\alpha_1} - 1)\|\nabla L_n(\mathbf{0})\|_2(1 - e^{-t_k})}{\|\theta(t_k)\|_2/(1 - 2^{-1})} \\
&\geq \frac{e^{t_k/2}(e^{\alpha_1} - 1)\|\nabla L_n(\mathbf{0})\|_2(1 - e^{-t_k})}{2(e^{t_k} - 1)\|\nabla L_n(\mathbf{0})\|_2} \geq 2^{-1}(e^{\alpha_1} - 1)e^{-t_k/2},
\end{aligned}
$$

which implies that

$$
A_k \geq \ln(1 + 2^{-1}(e^{\alpha_1} - 1)e^{-t_k/2}) \geq \frac{(e^{\alpha_1} - 1)e^{-t_k/2}}{2 + (e^{\alpha_1} - 1)e^{-t_k/2}} = \frac{(e^{\alpha_1} - 1)}{2e^{t_k/2} + (e^{\alpha_1} - 1)}.
\tag{S27}
$$

Thus,

$$
\alpha_{k+1} \geq \min\left(\alpha_{\max}, 2\alpha_k, \frac{(e^{\alpha_1} - 1)}{2e^{t_k/2} + (e^{\alpha_1} - 1)}\right).
\tag{S28}
$$

Now we prove the divergence of $t_k$ by contradiction. Suppose that $t_k$ does not diverge. Then there must exist a constant $T$ such that $t_k < T$ for all $k$. However, now we have

$$\alpha_{k+1} \geq \min\left(\alpha_{\max}, 2\alpha_k, \frac{(e^{\alpha_1} - 1)}{2e^{T/2} + (e^{\alpha_1} - 1)}\right),$$

which implies that $\alpha_k$ is lower bounded by a positive constant when $k$ is large enough, implying that $t_k$ should diverge. This is a contradiction. Hence, $t_k$ diverges.

Now we are ready to show that the algorithm must terminate in a finite number of iterations. If $t_{\max} < \infty$, then $t_k \geq t_{\max}$ must hold for large enough $k$ as $t_k$ diverges. If $t_{\max} = \infty$, then we have that $\theta(t_{\max}) = \theta^\star$ is finite by assumption. Therefore, the first termination criterion in (14) should also be met when $N$ is large enough because $t_N$ diverges.

Lastly, we prove (15) when the algorithm is terminated. Using (9) and (10) in Theorem 2, and applying inequality (S7) in Lemma S2, we have that

$$\|\theta(t_k)\|_2 \leq \left(1 + C_0 \frac{(e^{\alpha_k} - 1)}{(1 - e^{-t_k})}\right)\|\theta_k\|_2 \leq (1 + C_0 e^{\alpha_k})\|\theta_k\|_2 \leq 2\|\theta_k\|_2,$$

and

$$e^{-t_1}\left(\frac{e^{\alpha_1} - 1}{1 - e^{-t_1}}\right)^2 \|\theta(t_1)\|_2^2 \leq e^{\alpha_1}(e^{\alpha_1} - 1)^2 \|\nabla L_n(\mathbf{0})\|_2^2 \lesssim \epsilon,$$

$$e^{-t_{k+1}}\left(\frac{e^{\alpha_{k+1}} - 1}{1 - e^{-t_{k+1}}}\right)^2 \|\theta(t_{k+1})\|_2^2 \leq e^{-t_{k+1}}\left(\frac{e^{\alpha_{k+1}} - 1}{1 - e^{-t_{k+1}}}\right)^2 \frac{\|\theta(t_k)\|_2^2 (e^{t_{k+1}} - 1)^2}{(e^{t_k} - 1)^2}$$

$$= e^{\alpha_{k+1}} e^{-t_k}\left(\frac{e^{\alpha_{k+1}} - 1}{1 - e^{-t_k}}\right)^2 \|\theta(t_k)\|_2^2 \lesssim e^{-t_k}\left(\frac{e^{\alpha_{k+1}} - 1}{1 - e^{-t_k}}\right)^2 \|\theta_k\|_2^2$$

$$\leq c_1^2 (e^{\alpha_1} - 1)^2 \|\nabla L_n(\mathbf{0})\|_2^2 \lesssim \epsilon$$

for any $k \geq 1$ when the algorithm is terminated.

Now at termination, one of two conditions in (14) must be met. If $t_N \geq t_{\max}$ is met, then applying (10) in Theorem 2 and the above two bounds, it follows that

$$\sup_{0 \leq t \leq t_{\max}} \left\{f_t(\tilde{\theta}(t)) - f_t(\theta(t))\right\} \lesssim 4\left(1 + C_0^2 (e^{\alpha_{\max}} + e^{\alpha_{\max}/2} + 1)^2\right)\epsilon \lesssim \epsilon. \tag{S29}$$

If it is the other termination criterion $c_2 (e^{t_N} - 1)^{-1}\left(1 - e^{-\max(\alpha_N, t_{\max} - t_N)}\right) \leq \epsilon$ in (14) that is met first, then we must have $t_N < t_{\max}$ at termination, and

$$\frac{1 - e^{-\max(\alpha_N, t_{\max} - t_N)}}{e^{t_N} - 1} \max\left(\|\theta_N\|_2^2, \|\theta(t_{\max})\|_2^2\right)$$

$$\leq \frac{1 - e^{-\max(\alpha_N, t_{\max} - t_N)}}{e^{t_N} - 1} \max\left(4\|\theta(t_N)\|_2^2, \|\theta(t_{\max})\|_2^2\right) \leq \frac{4\epsilon}{c_2}\|\theta(t_{\max})\|_2^2 \lesssim \epsilon$$

where again we have used $\|\theta_N\|_2 \leq 2\|\theta(t_N)\|_2$ and the fact that $\|\theta(t_N)\|_2 \leq \|\theta(t_{\max})\|_2 \leq \mathcal{O}(1)$.

Using this and applying (9) in Theorem 2, we still have

$$\sup_{0 \leq t \leq t_{\max}} \left\{f_t(\tilde{\theta}(t)) - f_t(\theta(t))\right\} \lesssim \epsilon.$$

This completes the proof of Theorem 3. ∎

# 9 Proof of Theorem 4

*Proof of Theorem 4.* We first consider the case where $t_{\max} = \infty$. In view of the termination criterion (14), the algorithm will be terminated when $t_k = \mathcal{O}(\ln(\epsilon^{-1}))$. We define $N = \max\{k : t_k \leq \ln(\epsilon^{-1})\}$. By applying Lemma S1 and S2, we have

$$\frac{\|\theta_k\|_2}{1 - e^{-t_k}} \leq \frac{1}{1 - C_0 e^{\alpha_k}} \frac{\|\theta(t_k)\|_2}{1 - e^{-t_k}} \leq \frac{\|\theta(t_{\max})\|_2 + \|\nabla L_n(\mathbf{0})\|_2}{1 - C_0 e^{\alpha_k}}. \tag{S30}$$

Therefore, we have

$$\frac{e^{t_k/2}(e^{\alpha_1}-1)\|\nabla L_n(\mathbf{0})\|_2(1-e^{-t_k})}{\|\theta_k\|_2}$$

$$\geq \frac{e^{t_k/2}(e^{\alpha_1}-1)\|\nabla L_n(\mathbf{0})\|_2}{2(\|\theta(t_{\max})\|_2+\|\nabla L_n(\mathbf{0})\|_2)} \equiv \nu_1 e^{t_k/2}(e^{\alpha_1}-1)\,,$$

where we have used $C_0 e^{\alpha_k} \leq 2^{-1}$, and

$$\nu_1 = \frac{\|\nabla L_n(\mathbf{0})\|_2}{2(\|\theta(t_{\max})\|_2+\|\nabla L_n(\mathbf{0})\|_2)} < 1$$

will be treated as a problem-dependent constant. Using this, we obtain that

$$\alpha_{k+1} \geq \min\left\{2\alpha_k,\, \ln(1+\nu_1 e^{t_k/2}(e^{\alpha_1}-1))\right\}$$

when $\alpha_{k+1} < \alpha_{\max}$. Now we use induction to prove that $\alpha_{k+1} \geq \ln(1+\nu_1 e^{t_k/2}(e^{\alpha_1}-1))$. First, note that $\nu_1 < 1$ and we have that $\alpha_1 \geq \ln(1+\nu_1(e^{\alpha_1}-1))$. Suppose that this is true for $k-1$. Then, we have that

$$e^{2\alpha_k}-1 \geq e^{\alpha_k/2}(e^{\alpha_k}-1) \geq e^{\alpha_k/2}\nu_1 e^{t_{k-1}/2}(e^{\alpha_1}-1) \geq \nu_1 e^{t_k/2}(e^{\alpha_1}-1)\,.$$

Therefore, $2\alpha_k \geq \ln(1+\nu_1 e^{t_k/2}(e^{\alpha_1}-1)$ and

$$\alpha_{k+1} \geq \min\left\{2\alpha_k,\, \ln(1+\nu_1 e^{t_k/2}(e^{\alpha_1}-1))\right\} \geq \ln(1+\nu_1 e^{t_k/2}(e^{\alpha_1}-1)) \qquad \text{(S31)}$$

when $\alpha_{k+1} \leq \alpha_{\max}$.

Now, applying Lemma S3, it follows that the number of grid points $N$ satisfies

$$N \lesssim \epsilon^{-1/2} + \ln(\epsilon^{-1})/\alpha_{\max} = \mathcal{O}(\epsilon^{-1/2})\,,$$

if we treat $\beta$, $\|\theta(t_{\max})\|_2$, and $\|\nabla L_n(\mathbf{0})\|_2$ as constants.

When $t_{\max} < \infty$, we know that the algorithm will terminate if $t_N \geq t_{\max}$. Since $\alpha_k$ is lower bounded by an increasing sequence (S31), it follows that after at most $t_{\max}/\alpha_1 \lesssim \epsilon^{-1/2}$ iterations, we must have that $t_N \geq t_{\max}$ and the algorithm terminates. This completes the proof of Theorem 4. ∎

## 10 Proof of Theorem 5

*Proof of Theorem 5.* For any $t_k \leq t \leq t_{k+1}$ with $k \geq 1$, we have that

$$f_t(\tilde{\theta}(t)) - f_t(\theta(t)) = f_t(\theta_k) - f_t(\theta(t)) \leq \frac{1}{2\lambda(t)}\|\nabla f_t(\theta_k)\|_2^2$$

$$= \frac{1}{2\lambda(t)}\left\|\frac{1-e^{-t}}{1-e^{-t_k}}\nabla f_{t_k}(\theta_k) + \frac{e^{-t}-e^{-t_k}}{1-e^{-t_k}}\theta_k\right\|_2^2$$

$$\leq \frac{1}{\lambda(t)}\left\{\left(\frac{1-e^{-t}}{1-e^{-t_k}}\right)^2\|g_k\|_2^2 + \left(\frac{e^{-t}-e^{-t_k}}{1-e^{-t_k}}\right)^2\|\theta_k\|_2^2\right\}$$

$$\leq \frac{1}{\lambda_k}\left\{\left(\frac{1-e^{-t_{k+1}}}{1-e^{-t_k}}\right)^2\|g_k\|_2^2 + \left(\frac{e^{-t_{k+1}}-e^{-t_k}}{1-e^{-t_k}}\right)^2\|\theta_k\|_2^2\right\}\,, \qquad \text{(S32)}$$

Now using the bound in (18) and the fact that $\alpha_{k+1} \leq 2\alpha_k$ and $(1-e^{-t_{k+1}})/(1-e^{-t_k}) \leq 3$, we obtain that

$$f_t(\tilde{\theta}(t)) - f_t(\theta(t))$$

$$\leq \frac{1}{\lambda_k}\left\{\left(\frac{1-e^{-t_{k+1}}}{1-e^{-t_k}}\right)^2\left(C_0\frac{e^{\alpha_k}-1}{e^{t_{k-1}}-1}\|\theta_{k-1}\|_2\right)^2 + \left(\frac{e^{-t_{k+1}}-e^{-t_k}}{1-e^{-t_k}}\right)^2\|\theta_k\|_2^2\right\}$$

$$\leq \frac{2}{\lambda_k}\max_{j\in\{k,k+1\}}\left\{\frac{(e^{\alpha_j}-1)^2}{(e^{t_{j-1}}-1)^2}\|\theta_{j-1}\|_2^2\right\}\,,$$

for any $C_0 \le 3^{-1}$ and $k \ge 2$; and

$$
\begin{aligned}
&f_t(\tilde{\theta}(t)) - f_t(\theta(t)) \\
&\le \frac{1}{\lambda_k} \left\{ \left( \frac{1 - e^{-t_{k+1}}}{1 - e^{-t_k}} \right)^2 (C_0(e^{\alpha_1} - 1)\|\nabla L_n(\mathbf{0})\|_2)^2 + \left( \frac{e^{-t_{k+1}} - e^{-t_k}}{1 - e^{-t_k}} \right)^2 \|\theta_k\|_2^2 \right\} \\
&\le \frac{2}{\lambda_1} \max \left\{ \frac{(e^{\alpha_2} - 1)^2}{(e^{t_1} - 1)^2} \|\theta_1\|_2^2, \; (e^{\alpha_1} - 1)^2 \|\nabla L_n(\mathbf{0})\|_2^2 \right\},
\end{aligned}
$$

for any $C_0 \le 1/3$ when $k = 1$. Moreover, for any $0 \le t < t_1$, we have that

$$
f_t(\tilde{\theta}(t)) - f_t(\theta(t)) = f_t(\mathbf{0}) - f_t(\theta(t)) \le \frac{(1 - e^{-t})^2}{2\lambda(t)} \|\nabla L_n(\mathbf{0})\|_2^2 \le \frac{(1 - e^{-t_1})^2}{2\lambda_0} \|\nabla L_n(\mathbf{0})\|_2^2.
$$

$$(S33)$$

Now we bound $f_t(\tilde{\theta}(t)) - f_t(\theta(t))$ when $t_N < t \le t_{\max}$. Toward this end, notice that

$$
\begin{aligned}
&f_t(\tilde{\theta}(t)) - f_t(\theta(t)) = f_t(\theta_N) - f_t(\theta(t)) \\
&= \frac{1 - e^{-t}}{1 - e^{-t_N}} (f_{t_N}(\theta_N) - f_{t_N}(\theta(t_N))) + \frac{e^{-t_N} - e^{-t}}{2(1 - e^{-t_N})} (\|\theta(t_N)\|_2^2 - \|\theta_N\|_2^2) + f_t(\theta(t_N)) - f_t(\theta(t)).
\end{aligned}
$$

Next, we bound these three terms separately. For the first term, by using (S2), we have that $f_{t_N}(\theta_N) - f_{t_N}(\theta(t_N)) \le 2^{-1} e^{t_N} \|g_N\|_2^2$. Using this, we obtain that

$$
\frac{1 - e^{-t}}{1 - e^{-t_N}} (f_{t_N}(\theta_N) - f_{t_N}(\theta(t_N))) \le \frac{(1 - e^{-t})}{2\lambda(t_N)(1 - e^{-t_N})} \|g_N\|_2^2.
$$

$$(S34)$$

For the third term, similar to the proof of Theorem 1, we have that

$$
f_t(\theta(t_N)) - f_t(\theta(t)) \le \frac{e^{-t_N} - e^{-t}}{2(1 - e^{-t_N})} \left( \|\theta(t)\|_2^2 - \|\theta(t_N)\|_2^2 \right).
$$

Combining the bounds for the first term and the third term, and the bound in (18), we obtain that

$$
\begin{aligned}
f_t(\tilde{\theta}(t)) - f_t(\theta(t)) &\le \frac{(1 - e^{-t})}{2\lambda(t_N)(1 - e^{-t_N})} \|g_N\|_2^2 + \frac{e^{-t_N} - e^{-t}}{2(1 - e^{-t_N})} (\|\theta(t)\|_2^2 - \|\theta_N\|_2^2) \\
&\le \frac{(1 - e^{-t})}{2\lambda(t_N)(1 - e^{-t_N})} \left( C_0 \frac{e^{\alpha_N} - 1}{e^{t_{N-1}} - 1} \|\theta_{N-1}\|_2 \right)^2 + \frac{e^{-t_N} - e^{-t}}{2(1 - e^{-t_N})} (\|\theta(t)\|_2^2 - \|\theta_N\|_2^2) \\
&\le \frac{1}{\lambda(t_N)} \frac{(e^{\alpha_N} - 1)^2}{(e^{t_{N-1}} - 1)^2} \|\theta_{N-1}\|_2^2 + \frac{\|\theta(t)\|_2^2}{2(e^{t_N} - 1)}
\end{aligned}
$$

for any $t_N < t \le t_{\max}$, provided that $e^{-t_N} \le 17/18$ and $C_0 \le 1/3$.

Now, combining the bounds for $f_t(\tilde{\theta}(t)) - f_t(\theta(t))$ when $t \in [t_k, t_{k+1}]$ for $0 \le k \le N$, we have that

$$
\sup_{t \in [0, t_{\max}]} f_t(\tilde{\theta}(t)) - f_t(\theta(t)) \le 2 \max \left\{ \frac{(e^{\alpha_1} - 1)^2 \|\nabla L_n(\mathbf{0})\|_2^2}{\min(\lambda_0, \lambda_1)}, \; \max_{1 \le k \le N-1} \frac{(e^{\alpha_{k+1}} - 1)^2 \|\theta_k\|_2^2}{(e^{t_k} - 1)^2 \min(\lambda_k, \lambda_{k+1})} \right\},
$$

when $t_{N-1} < t_{\max} < t_N$, and

$$
\begin{aligned}
&\sup_{t \in [0, t_{\max}]} f_t(\tilde{\theta}(t)) - f_t(\theta(t)) \\
&\le \; 2 \max \left\{ \frac{(e^{\alpha_1} - 1)^2 \|\nabla L_n(\mathbf{0})\|_2^2}{\min(\lambda_0, \lambda_1)}, \; \max_{1 \le k \le N-1} \frac{(e^{\alpha_{k+1}} - 1)^2 \|\theta_k\|_2^2}{(e^{t_k} - 1)^2 \min(\lambda_k, \lambda_{k+1})}, \; \frac{\sup_{t \ge t_N} \|\theta_t\|_2^2}{e^{t_N} - 1} \right\},
\end{aligned}
$$

when $t_N < t_{\max}$, provided that $\alpha_{k+1} \le 2\alpha_k$ and $t_N \ge \ln(18/17)$. Using the step size scheme in (19) and (20), we obtain that

$$
\sup_{t \in [0, t_{\max}]} f_t(\tilde{\theta}(t)) - f_t(\theta(t)) \le 2\epsilon \max \left( 1, \sup_{t \ge t_N} \|\theta_t\|_2^2 / c_2 \right) \lesssim \epsilon
$$

$$(S35)$$

when the algorithm is terminated for some constant $c_2 \geq 1$.

Finally, it remains to show that the algorithm must terminate within a finite number of steps. To this end, suppose that $t_k$ does not diverge. Then there exists constant $\bar{t} < \infty$ such that $t_k \leq \bar{t}$ for all $k \geq 1$. Using Assumption (A1), we have that $\lambda_k \geq \min_{t_k < t < t_{k+1}} g(t)$. Therefore,

$$\min(\lambda_k, \lambda_{k+1}) \geq \min_{t < t_{k+2}} g(t) \geq g(\bar{t}) . \tag{S36}$$

Moreover, using $\|g_k\|_2 \leq 1$ and

$$f_{t_k}(\theta_k) - f_{t_k}(\mathbf{0}) \leq f_{t_k}(\theta_k) - f_{t_k}(\theta(t_k)) \leq \frac{1}{2\lambda(t_k)}\|g_k\|_2^2 \leq \frac{1}{2\lambda(t_k)} , \tag{S37}$$

we obtain that

$$\|\theta_k\|_2^2 \leq \frac{e^{t_k}}{\lambda(t_k)} + 2(e^{t_k} - 1)(L_n(\mathbf{0}) - L_n(\theta_k)) \leq \frac{e^{\bar{t}}}{g(\bar{t})} + 2(e^{\bar{t}} - 1)(L_n(\mathbf{0}) - M) \tag{S38}$$

where $M = \inf_\theta L_n(\theta) > -\infty$. Combining these bounds, we get

$$\alpha_{k+1} = \min\left\{\alpha_{\max}, 2\alpha_k, \ln\left(1 + \frac{c_1(e^{\alpha_1} - 1)\|\nabla L_n(\mathbf{0})\|_2(e^{t_k} - 1)\sqrt{\min(\lambda_k, \lambda_{k+1})}}{\|\theta_k\|_2\sqrt{\min(\lambda_0, \lambda_1)}}\right)\right\}$$
$$\geq \min\left\{\alpha_{\max}, 2\alpha_k, \mathcal{O}(1)\right\} ,$$

which implies that $\alpha_{k+1}$ is lower bounded by a constant, which contradicts with the fact that $t_k \leq \bar{t}$ for all $k \geq 1$. Hence, $t_k$ must diverge, and the algorithm must terminate after a finite number of steps. This completes the proof of Theorem 5. $\blacksquare$