# OpenReview forum: "Path following algorithms for $\ell_2$-regularized $M$-estimation with approximation guarantee"
_NeurIPS.cc/2023/Conference — NeurIPS 2023 poster_

### Official Review · Reviewer_toMd · 2023-07-04

**Soundness:** 2 fair
**Presentation:** 1 poor
**Contribution:** 1 poor
**Rating:** 4
**Confidence:** 2

**Summary:**

A novel grid point selection scheme and stopping criterion for any general path-following algorithms are proposed in this paper. However, the presentation of this paper is poor, and the contributions are not clear.

**Strengths:**

A novel grid point selection scheme and stopping criterion for any general path-following algorithms are proposed in this paper.

This work provides some interesting theoretical results. For example, this work theoretically shows that the approximation error of the proposed solution path can be upper bounded by the sum of the interpolation and optimization error, and establish a global approximation-error bound for the solution path.

**Weaknesses:**

This work provides some interesting theoretical results, however, I am dissatisfied with the writing and organization of this paper.

1. The writing of this paper is poor. In order to enhance the readability, I suggest the author discuss the definition of "regularized M-minimization problems" in the introduction, maybe with some formulation.

2. The contributions of this paper are not clear. This work shows some interesting theoretical results,  I suggest the author clearly clarify the contributions in the Introduction.

3. The experimental results are all put in the Appendix in the current manuscript. I am unsure why the authors did not put the results in the main text, especially considering that there are many empty spaces on the ninth page of the main text.

4. The format of the appendix may not comply with the requirements of NeurIPS. I recommend that the authors make corrections accordingly.

5. To further improve the readability of this paper, I suggest the author add a "Related Work" section to introduce some works related to this work. And the organization of this work is poor, I suggest the author carefully modify the whole paper.



**Questions:**

1. The writing of this paper needs improvement. It is suggested to enhance readability by discussing the definition of "regularized M-minimization problems" in the introduction, possibly with some formulation.

2. Lack of clarity regarding the contributions of this paper. Although this work presents some interesting theoretical results, it is recommended that the author explicitly clarify the contributions in the Introduction.

3. Placement of experimental results in the Appendix. Uncertain why the authors chose not to include the results in the main text, especially considering the available space on the ninth page.

4. Non-compliance of the appendix format with NeurIPS requirements. It is advised that the authors make the necessary corrections to adhere to the guidelines.

5. Readability and organization concerns. To enhance the paper's readability, it is suggested to add a "Related Work" section to introduce works relevant to this study. Furthermore, the entire paper's structure requires careful modification.

---

> ### Author Rebuttal · Authors · 2023-08-10
>
> Thanks for your valuable comments and suggestions.
> We summarize our responses below.
>
> **Weaknesses and Questions**
>
> 1. Thanks for the suggestion. Following your suggestion,
> we introduced the definition of regularized $M$-estimation
> earlier in the Introduction section in our revision.
>
> 2. Thanks for the suggestion. We have made a few changes to the Introduction section to better highlight our contributions.
> In particular, the key contributions are summarized in the third and fourth paragraphs in Section 1.
>
> 3. Thanks for the suggestion. We now include the *runtime versus global approximation error* plot for the simulation studies  (Figure S1 and Figure S3 of current supplementary file) in the main file. We may move additional plots in the Appendix to the main text if the paper is accepted.
>
> 4. Thanks for the suggestion. We now have changed the format of the supplementary material by using the NeurIPS template.
>
> 5. Thanks for the suggestion. Most of the discussions of related work can be found in the last few paragraphs of the Instroduction section starting from line 62 of the current main text file.
> The discussions involve summarizing relevant work as well as highlighting their differences and new contributions of our work
> compared to existing works.
> As such, we feel that the contributions of the article should be quite clear to the reader, and a separate Related Work subsection may not be necessary.

---

> > ### Comment · Reviewer_toMd · 2023-08-21
> >
> > Thank you for the response. I tend to maintain my scores.

---

### Official Review · Reviewer_omjq · 2023-07-06

**Soundness:** 3 good
**Presentation:** 1 poor
**Contribution:** 2 fair
**Rating:** 6
**Confidence:** 3

**Summary:**

This paper proposes a new path following algorithm for 2-regularized M-estimation with an approximation guarantee. The algorithm includes a grid point selection scheme and an adaptive stopping criterion to optimize the trade-off between model fit and complexity in machine learning algorithms. The paper also provides a comparison of the proposed scheme to a standard path-following scheme through a simulated study using ridge regression and L2-regularized logistic regression.

**Strengths:**

1. This paper proposes a new path following algorithm for 2-regularized M-estimation with an approximation guarantee.

2. This work is generally technically sound with rigorous statistical analysis in the work.

3. The article is well written and mathematical derivations are easy to follow.

**Weaknesses:**

1: The main body does not include the experimental results.

2: The experiments were solely conducted on simulation datasets, and evaluating the method on public datasets could enhance the results.

3: The experiments do not include other baselines on solution path algorithms.

4: Can the proposed method be extended to other regularized M-estimation techniques, such as Lasso or Group Lasso?

5: It is important to discuss relevant existing work on solution path algorithms, including notable studies such as [1], as well as approximate solution path algorithms, like [2].

[1] Ryan Joseph Tibshirani, Jonathan E Taylor, Emmanuel Jean Candes, and Trevor Hastie, "The solution path of the generalized lasso," The Annals of Statistics, 2011.

[2] Runxue Bao, Bin Gu, and Heng Huang, "Efficient Approximate Solution Path Algorithm for Ordered Weighted L_1-Norm with Accuracy Guarantee," ICDM 2019.

**Questions:**

see weaknesses

---

> ### Author Rebuttal · Authors · 2023-08-10
>
> Thanks for your valuable comments and suggestions.
> We summarize our responses below:
>
> **Weaknesses and Questions**
> 1. Thanks for the suggestion. We have moved all the *runtime versus global approximation error* plot for the simulation studies (Figure S1 and Figure S3 of current supplementary file) to the main text. We may move additional plots in the Appendix to the main text if the paper is accepted.
>
> 2. Thanks for your suggestions. We have added a real-world data example in the revision (please see plots in Section 1 of our author rebuttal pdf).
>
> 3. Thanks for the suggestion. Our current baseline algorithm selects equally spaced grid points (on a log-scale), which is widely adopted by many popular packages such as glmnet.
>
> 4. Unfortunately, it seems that our proposed method can only be easily extended to the differentiable regularizer. Extensions to nonsmooth regularizer seem to be not straightforward, and will be left for future work.
>
> 5. Thanks for pointing out additional related works. We have included them and some discussions in the revision.

---

> > ### Comment · Reviewer_omjq · 2023-08-13
> >
> > Thank you very much for the response.

---

> > > ### Author Response · Authors · 2023-08-13
> > >
> > > Thanks for the reply. Kindly let us know if your concerns have been fully resolved. We will also be very happy to answer any remaining questions you may have.

---

> > > ### Author Response · Authors · 2023-08-14
> > > **Does our response address your questions?**
> > >
> > > Thanks again for the efforts and time you put into reading and commenting on our work. We wonder whether our response has addressed your concerns and questions. We would appreciate the opportunity to engage further if needed. We also kindly ask you to consider stronger support for the paper if your concerns have been addressed.
> > >
> > > Thanks!

---

### Official Review · Reviewer_5bG4 · 2023-07-06

**Soundness:** 3 good
**Presentation:** 3 good
**Contribution:** 3 good
**Rating:** 6
**Confidence:** 3

**Summary:**

This paper considers this problem of the ($\ell_2$-regularized) $M$-estimation problem of computing solutions to the regularized objective $(e^t – 1) L(\Theta) + (1/2) \| \Theta \|_2^2$ for all $t$, denoted $\theta(t)$. The paper proposes a method for approximating theta(t) for carefully chosen values of t, such that interpolating between these points approximately solves the problem. When $L$ is convex and differentiable the paper proves that (under certain bounds) $1/\sqrt{\epsilon}$ values of t suffice to obtain a curve of $\epsilon$ approximate solution to the regularized objective. The paper discusses how this improves upon prior bounds, how the scheme has desirable properties for implementation, and provides experiments implementing the algorithm. Further, the paper discusses extensions of their work to non-convex optimization.

**Strengths:**

The $M$-estimation problem that the paper considers is a natural and feels fundamental. This paper provides a seemingly useful set of theoretical (algorithms and bounds) and empirical (experiments) on this fundamental problem. Further, the paper argues how they improve upon prior work on this problem. Additionally, the paper is fairly well-written, suggests an interesting direction or future work, and could invite further study of the problem.

**Weaknesses:**

The problem considered and the approach for it feels quite natural. This is not necessarily an issue except that it isn’t clear from the writing if the results are not what one might obtain by following the first approaches for the problem one might think of. For example, if one simply computes the derivative of the value of the regularized objective minimizer with respect to t and then bounds the terms, does that more or less directly suggest the approach of the paper? More broadly, I think the paper could benefit by discussing their proof strategy and any obstacles that had to be overcome in achieving their result.

Additionally, I thought it would be helpful if the paper was a little clearer about what assumptions need to be made to achieve, e.g., the claimed $O(1/\sqrt{\epsilon})$. What other quantities does this rate depend on exactly? As discussed briefly in the following “questions” section I think it would be helpful to discuss all the quantities that the proposed method depends on.

Finally, the $M$-estimation problem seems like one that could be studied in part under different names in a number of different areas. Ultimately, it is in some sense asking to approximately follow a certain induced curve which seems like something that could have arisen in a number of contexts. Consequently, more literature review or discussion to raise confidence in the novelty of the proposed method would be helpful.


**Questions:**

My main question are those raised in the “Weaknesses” category. Additionally, below are some more detailed suggestions, questions, and comments:

* Line 16: “Corroborate with our theoretical” --> “Corroborate our”
* Line 62 – 99: It would be helpful to know what exactly is hidden in the $O(\cdot)$ notation.
* Line 112: why write the $n$ in $L_n$? Does it arise anywhere else in the paper?
* Theorem 1: some more discussion of the proof strategy would be nice.


**Limitations:**

The paper doesn’t discuss weaknesses and limitations, but the paper is primarily bout the theoretical and empirical analysis of an algorithm for solving a sequence of regularized optimization problems. Consequently, the societal impact is unclear. Further, the weaknesses and questions discussed reflect limitations for which further discussion might be beneficial.

---

> ### Author Rebuttal · Authors · 2023-08-10
>
> Thanks for your valuable comments and suggestions.
> We summarize our responses below:
>
> **Weaknesses**
>
> * Thanks for the suggestion, we have added the proof strategy for Theorem 1. Basically, the proof of Theorem 1 starts with relating the local approximation error to the suboptimality of $\theta_k$ and $\theta_{k+1}$ at
> some $t \in [t_k, t_{k+1}]$. The suboptimality can then further bounded
> by the square norm of the grandient leveraging the fact that the objective function is
> $e^{-t}$-strongly convex. Finally, a triangular inequality is used
> to bound the norm of the grandient by quantities that depend on $\|g_k\|_2$ and $\|\theta_k\|_2$.
>
> * The hidden quantities behind our claimed number of iterations $\mathcal O(1/\sqrt{\epsilon})$ include
> parameters $c_2$, $\alpha_{\max}$, $t_{\max}$, as well as the problem-dependent constants $\|\theta(t_{max})\|_2$
> and $\|{\nabla} L_n(0)\|_2$.
>
> * Basically, a larger $c_2$ or $\alpha_{\max}$ will lead to fewer number of iterations, while a larger $t_{\max}$ will require more iterations.
> The problem-dependent constants $A = \|\theta(t_{\max})\|_2$ and $B = \|\nabla L_n(0)\|_2$ impact the number of iterations jointly through quantity $\nu_1 = B/(2(A+B))$. The number of iterations needed will increase as $\nu_1$ increase.
>
> * Thanks for the suggestion, we have included some additional related works and additinoal discussions in the Introduction section.
>
> **Questions**
> * Fixed.
> * See the response in above **Weaknesses** section.
> * Since the loss function is often formulated based on the training examples,
> we use notation $L_n$ to reflect that $n$ examples are considered when we define the loss function.
> For example, in the section 4.2 of our current main text file, we have that the empirical loss function for logistic regression is
> $L_n(\theta) = n^{-1} \sum_{i=1}^n \log(1+\exp(-Y_i X_i^\top\theta))$.
>
> * Thanks for the suggestion, we have added the proof strategy for Theorem 1 in the revision. The basic idea can be found in above **Weaknesses** section.

---

> > ### Comment · Reviewer_5bG4 · 2023-08-14
> > **Response to Rebuttal**
> >
> > Thank you or your response. I appreciate your answers and my overall evaluation remains as is.

---

> > > ### Author Response · Authors · 2023-08-14
> > > **Thanks for the reply**
> > >
> > > Thanks for the reply. If you have any further questions or concerns regarding our work, please don't hesitate to let us know.

---

### Official Review · Reviewer_2KPJ · 2023-07-08

**Soundness:** 2 fair
**Presentation:** 3 good
**Contribution:** 2 fair
**Rating:** 6
**Confidence:** 3

**Summary:**

The submission 9422 proposed a novel selection scheme for grid search and stopping criterion for a more general path-following algorithms, while not trying to compute/derive the whole solution spectrum. The analysis of the authors indicates that under certain assumptions, their proposed approximate path-tracking algorithm (with linear interpolation) can approximate the exact solution path and provide theoretical boundaries. The theoretical conclusions have been verified to some extent through some simple numerical simulations.

**Strengths:**

- Solution path (hyperparameter optimization) is an interesting and important topic in both theory and application communities of ML.

- The paper is well written and it clearly state its contributions, notation and results. The most part of their derivations are easy to follow (for me).

**Weaknesses:**

- The numerical simulations of this work is too weak and might not be convincing.
  - the scale of the experiment is too small. The authors are suggested to consider more objectives since their analysis can be readily extended to general differentiable functions
  - lack of experiments on real-world benchmark datasets, e.g., UCI, libsvm
  - the code (currently) has not been submitted and lacks *sufficient* details for reproducibility

- Lack of some existing research work on the situation where the exact paths are not piecewise linear, e.g.,

  - [1] Pierre Garrigues, and Laurent Ghaoui. *An homotopy algorithm for the Lasso with online observations.*  Advances in neural information processing systems 21 (2008).

  - [2] Xingyu Qu, Diyang Li, Xiaohan Zhao, and Bin Gu. *GAGA: Deciphering Age-path of Generalized Self-paced Regularizer.* Advances in Neural Information Processing Systems 35 (2022).

  - ...

I also suggest that the authors further highlight the intrinsic differences from recent work and the technical challenges faced by their settings.

**Questions:**

- Why not use the vanilla $C(t):=t$ to replace the current $C(t):=e^t-1$ in your analysis?

- I suggest the authors to plot the *path visualization* of $\tilde{\theta}(t)$ as well as $\theta(t)$, to help the readers that may not familiar with the solution path area. also we can observe how the computed path (via algo. 1) approximates the ground truth.

**Limitations:**

- some technical limitations of the current method were mentioned in some parts of the paper, but there was no detailed/thorough discussion.

- there is no need to discuss potential negative social impacts.

---

> ### Author Rebuttal · Authors · 2023-08-10
>
> Thanks for your valuable comments and suggestions.
> We summarize our response as follows.
>
> **Weaknesses**
>
> * Thanks for your suggestions on numerical simulations. Following your suggestion, we have added a real-world data example in our numerical studies section (please see plots in Section 1 of our author rebuttal pdf).  We will make all the code publicly available if the article gets accepted.
> * Thanks for pointing out additional related works. We have included those works and added some additional discussions in our revision.
>
> **Questions**
> * We agree that our current choice of $C(t) = e^t - 1$ is not essential
> and can be replaced by $C(t) = t$.
> The primary rationale behind choosing $C(t) = e^t - 1$ is that grid points selection are typically equally spaced in the log-scale.
> Moreover, it makes some of the algebra much simpler and cleaner in our theoretical derivations.
>
> * Thanks for the suggestion. We have included the visualization of $\tilde{\theta}(t)$ and $\theta(t)$ of simple examples in the supplementary material in the revision (please see plots in Section 2 of our author rebuttal pdf). We will move this part to the main text if the article gets accepted.

---

> > ### Comment · Reviewer_2KPJ · 2023-08-14
> > **Thank you for your explanation and additional numerical simulations...**
> >
> > I also have experience running solution paths algo on datasets such as *a9a*. I hope the author could provide more real-world datasets in the final version, which may make your results more convincing.
> >
> > In addition, in Figure S2, the ground truth and the method proposed in this work almost overlap, which may not be very convincing (because if I simply ran the same algorithm twice, it would also have the same Figures). It is best to change some settings/conditions to clearly demonstrate how the solution path algo approaches the ground truth.
> >
> > Given the overall reviews and rebuttal, I think this work could be a good contribution, and will upgrade my score 5 -> 6

---

> > > ### Author Response · Authors · 2023-08-14
> > > **Thank you for upgrading your score.**
> > >
> > > Thanks for the additional suggestions! We will provide results on more real-world datasets in our final version, and present new plots on $\theta(t)$ versus $\tilde{\theta}(t)$ to more clearly demonstrate how our solution path algo approaches the ground truth. If you have any further questions or concerns regarding our work, please don't hesitate to let us know.

---

### Author Rebuttal · Authors · 2023-08-10

Thanks for all of your valuable comments and suggestions.
Following your suggestions,
we have made some changes to our submission, which we feel greatly improved the presentation.
Main changes include:

* We moved the formal definition of $M$-estimation earlier in the Introduction section, and made a few changes to the Introduction section to better highlight our contributions.
* We included suggested related literature and added additional discussions of the existing works.
* We moved the *runtime versus global approximation error* plots of our numerical studies into the main file.
* We added plots to visualize our approximated solution path as well as the ground truth solution in the Appendix (also see plots in Section 2 of our author rebuttal pdf). We will include them in the main file if our paper gets accepted.
* We added a real data analysis example in Section 4 (also see plots in Section 1 of our author rebuttal pdf). We will place the entire real data analysis section in the allowed additional content page if our paper gets accepted.

---

### Author Response · Authors · 2023-08-10
**Happy to Respond to Any Further Comments/Clarifications**

Dear Reviewers,

Thanks for the effort you put into reading and commenting on our work. Currently, you have given our work scores: 5, 6, 5 and 4, and have described our paper positively in the following way:

* Solution path (hyperparameter optimization) is an interesting and important topic in both theory and application communities of ML.

* This paper provides a seemingly useful set of theoretical (algorithms and bounds) and empirical (experiments) on this fundamental problem, and is generally technically sound with rigorous statistical analysis.

* The paper is fairly well-written, suggests an interesting direction or future work, and could invite further study of the problem.

You have also raised some questions/concerns, to which we have replied in our detailed rebuttal. Specifically, we

* Moved the formal definition of $M$-estimation earlier in the Introduction section to enhance the readability, and made a few changes to the Introduction section to better highlight our contributions.

* Included more related literature and added additional discussions of the existing works.

* Moved the plots of our numerical studies from the appendix into the main file.

* Added a real data analysis example in Section 4 to enhance our result (also see plots in Section 1 of our author rebuttal pdf).

* Added plots to visualize our approximated solution path as well as the ground truth solution (also see plots in Section 2 of our author rebuttal pdf).

* Added discussions on the proof strategy of Theorem 1.

* Changed the format of the supplementary material by using the NeurIPS template.

The additional experiments plots have also been carried out, please check them in the attached PDF of our author rebuttal. We also kindly request you to consider stronger support for the paper if your concerns have been addressed. Please feel free to let us know if you still have any remaining concern, we will be happy to address them.

Thanks for your time and support!

Authors of Paper #9422

---

### Decision · Program_Chairs · 2023-09-21

**Decision:**

Accept (poster)

**Comment:**

I am recommending acceptance but strongly recommend that the authors take into account the comments of the reviewers. As examples of issues brought up by the reviewers:
1. The paper would be more informative with extended results on real data. Simulated data often has nice properties, and some methods that work on simulated data do not work well on real data.
2. Extended discussion of related works and what rates they would achieve, and comparison to suggested baselines in the experiments.

Another line of work that the authors may not be aware of are methods that search along the solution with the goal of finding a particular solution. These were previously popular for L1-regularization, with the two most-impactful works probably being:
- Probing the Pareto frontier for basis pursuit solutions
- A Proximal-Gradient Homotopy Method for the Sparse Least-Squares Problem